

# HONO Formation Mechanisms and Impacts on Ambient Oxidants in Coastal Regions of Fujian, China

Haoran Zhang[1], Chengchun Shi[2,3], Chuanyou Ying[1,4], Shengheng Weng[5], Erling Ni[2,3], Lanbu Zhao[1], Peiheng Yang[1], Keqin Tang[6], Xueyu Zhou[1], Chuanhua Ren[1], Tengyu Liu[1], Mengmeng Li[1], Nan Li[6], Xin Huang[1*]

[1] School of Atmospheric Sciences, Nanjing University, Nanjing, 210023, China
[2] Fujian Academy of Environmental Sciences, Fuzhou, 350013, China
[3] Fujian Key Laboratory of Environmental Engineering, Fuzhou, 350013, China
[4] Fuzhou Research Academy of Environmental Sciences, Fuzhou, 350013, China
[5] Fujian Institute of Meteorological Sciences, China Meteorological Administration, Fuzhou, 350001, China
[6] Jiangsu Key Laboratory of Atmospheric Environment Monitoring and Pollution Control, Jiangsu Collaborative Innovation Center of Atmospheric Environment and Equipment Technology, School of Environmental Science and Engineering, Nanjing University of Information Science & Technology, Nanjing, 210044, China

*Correspondence to*: Xin Huang (xinhuang@nju.edu.cn)

**Abstract.** Nitrous acid (HONO) is a vital precursor of hydroxyl radicals (OH) in the troposphere, leading to the formation of secondary air pollutants, including ozone ($O_3$) and secondary aerosols. Previous studies have mainly focused on investigating the chemical fate of HONO in polluted urban areas of China and found a general diurnal variation featuring the minimum concentration around noon. However, this study reported a significantly higher daytime HONO concentrations based on one-month measurement during May of 2024 over the coastal regions of Fujian in southeastern China. Using an updated chemical transport model, we captured the magnitude and temporal variation observed in coastal HONO levels, and improved the model performance on diurnal patterns of the $NO_2$ and $O_3$. Further process analysis revealed that two light-dependent chemical sources, i.e., the heterogeneous uptake of $NO_2$ on the ground surface and $NO_x$ photo-oxidation, were the main contributors to HONO formation, particularly at high concentrations around noon in the presence of persistent intensive solar radiation. In addition, we assessed that shipping emissions contributed 20% to the midday HONO production rate in coastal regions. Subsequently, model results indicated that HONO photolysis accounted for 34% of primary OH sources during the daytime. Model sensitivity experiments demonstrated that incorporating multiple HONO sources increased the daily maximum OH and average $O_3$ concentrations by 61% and 44%, respectively, in coastal regions. Overall, this study highlights the unique formation mechanisms of HONO and its significant contribution to ambient oxidants in typical coastal regions.

**Keywords.** Nitrous acid, Ozone, OH radicals, Southeast China, WRF-Chem



## 1 Introduction

Nitrous acid (HONO) is an unstable reactive nitrogen species. It can be generated through various formation pathways, such as direct emissions from combustion-related processes, soils, and fertilization (Kurtenbach et al., 2001; Oswald et

al., 2013; Su et al., 2011; Tan et al., 2023), the homogeneous reaction between NO and OH (R1) (Sarwar et al., 2008), light-enhanced $NO_2$ heterogeneous uptake on solid surfaces (R2) (Finlayson-Pitts et al., 2003; George et al., 2015; Kim et al., 2024), and photolysis of particulate nitrate aerosols (R3) (Ye et al., 2016, 2017). In addition, a recent laboratory study also reported that the photo- and dark-oxidation of nitrogen oxides ($NO_x$) could significantly contribute to HONO formation (Song et al., 2023).

$$NO + OH \rightarrow HONO \tag{R1}$$
$$2NO_2 + H_2O + hv \xrightarrow{Surface} HONO + HNO_3 \tag{R2}$$
$$pNO_3^- + hv \rightarrow 0.67\ HONO + 0.33\ NO_2 \tag{R3}$$

In the presence of sunlight, HONO can rapidly undergo photodissociation (R4) to yield NO and OH radicals (Seinfeld and Pandis, 2016). This photolytic process of HONO plays a vital role in maintaining the atmospheric oxidizing capacity (AOC) and facilitating the formation of secondary air pollutants (Kleffmann et al., 2005). For example, during the daytime, the oxidation of $NO_2$ by OH usually results in the production of nitric acid ($HNO_3$) (R5), which subsequently reacts with ammonia ($NH_3$) to form inorganic nitrate aerosols (R6) (Zhang et al., 2024a). Adaptationally, OH radicals can

degrade organic volatile organic compounds (VOCs), leading to the formation of $RO_2$ radicals (R7) that further contribute to the generation of secondary organic aerosols (SOA) (Wang et al., 2017). Concurrently, $RO_2$ can react with NO to produce $NO_2$ (R8), a reaction that reduces ozone ($O_3$) titration (R9) while providing $NO_2$ for the subsequent formation of $O_3$ through reactions R10 and R11.

$$HONO \rightarrow OH + NO \tag{R4}$$
$$OH + NO_2 \rightarrow HNO_3 \tag{R5}$$
$$HNO_3 + NH_3 \rightarrow NH_4NO_3 \tag{R6}$$
$$VOC + OH \rightarrow RO_2 + H_2O \tag{R7}$$
$$RO_2 + NO \rightarrow RO + NO_2 \tag{R8}$$
$$O_3 + NO \rightarrow O_2 + NO_2 \tag{R9}$$
$$NO_2 + hv \rightarrow O^3P + NO \tag{R10}$$
$$O^3P + O_2 \rightarrow O_3 \tag{R11}$$

Over the past decade, there has been a notable increase in research interests about the topics related to HONO due to its

critical potential to induce secondary air pollution in China (Jiang et al., 2024; Xue, 2022). The majority of previous studies have focused on elucidating the mechanisms of HONO formation and its associated environmental impacts, especially in city clusters of China that frequently experience severe air pollution (Fu et al., 2019; Li et al., 2022; Ran et al., 2024; Zhang et al., 2021, 2022c, 2023). For instance, Fu et al. (2019) demonstrated that HONO can intensify $O_3$





pollution by up to 24 ppbv during haze events in the Pearl River Delta (PRD) region of China. Meanwhile, Zhang et al.

(2021) concluded that the enhanced AOC conditions contributed by HONO chemistry increased secondary aerosol concentrations by 18–51% in the North China Plain (NCP). We also previously investigated the role of HONO in the synergistic evolution of particulate matter and $O_3$ compound air pollution over the Yangtze River Delta (YRD) region (Zhang et al., 2024b).

Observational studies conducted in polluted urban areas indicated that lower concentrations of HONO typically occurred

around noon (Fu et al., 2019; Song et al., 2023; Wang et al., 2025; Zhang et al., 2021). However, a recent in-situ measurement study in coastal regions revealed an inverse diurnal variation pattern of HONO concentrations. Specifically, Zhong et al. (2023) reported that their measurement campaign in Qingdao, a coastal city adjacent to the Yellow Sea, identified an unexpected diurnal peak in HONO concentrations at 12:00 local time (UTC+8). This suggests that oceanic contributions may affect the diurnal formation of HONO on land. But the mechanisms of HONO formation in coastal

regions are not well understood by existing studies. Moreover, the maritime industry is an essential source of oceanic emissions. A previous study confirmed that shipping activities can emit substantial amounts of $NO_x$ (Liu et al., 2016). As $NO_x$ is a key precursor of HONO, shipping emissions can considerably impact the subsequent chemical production of HONO on land near the ocean (Dai and Wang, 2021). Furthermore, HONO emissions have been acknowledged as a result of shipping activities (Ke et al., 2025; Sun et al., 2020). Nevertheless, the contribution of shipping emissions to

HONO formation in coastal areas has yet to be quantitatively assessed.

This study aims to investigate the mechanisms of HONO formation and its impact on the enhancement of OH radicals and $O_3$ in Fujian, a representative coastal region in southeastern China. To this end, we first conducted a one-month field measurement to characterize the levels and diurnal patterns of coastal HONO concentrations (see Section 3.1). We then employed the Weather Research and Forecasting model coupled with Chemistry (WRF-Chem) to reproduce the observed

variations in HONO and performed a chemical budget analysis of HONO using process analysis techniques (refer to Sections 3.2 and 3.3). The model utilized in this study is based on our previous updates of the multiple HONO sources (Zhang et al., 2024b). In Section 3.3, we conducted a sensitivity experiment by zeroing out shipping emissions to clarify the influence of maritime activities on coastal HONO formation. Subsequently, we assessed the contribution of HONO photolysis to elevated coastal OH radicals and $O_3$ concentrations, as described in Section 3.4. The relevant uncertainties

in the model were discussed in Section 3.5. Overall, this study comprehensively integrated field measurements and three-dimensional (3D) numerical simulations to improve our understanding of HONO formation mechanisms and its environmental implications for ambient oxidants in coastal regions of southeastern China.



## 2 Data and Methods

### 2.1 Field Measurements and Instrument

The in-situ measurement was conducted during the period from 1 to 31 May of 2024. The sampling location was the Dongzhen Reservoir (DZSK, 118.98°E, 25.48°N), which was categorized as a suburban site in Putian. The DZSK site was 6 km from downtown of Putian and approximately 25 km from the Taiwan Strait. The geographical location of DZSK is shown in Figure 1. Hourly HONO concentrations were measured using a long-path absorption photometer (LOPAP). The other two gaseous air pollutants, $NO_2$ and $O_3$, were measured using a commercial chemiluminescence

instrument (Thermo Fisher Scientific 17i and 49i). Air temperature and relative humidity were monitored by a micro automatic weather station (LUFFT). Ultraviolet radiation (UV-A) was simultaneously observed with a solar radiation instrument (Kipp & Zonen). As records of the hourly wind field and precipitation were missing for the study period, observations were obtained from alternative sources. For wind speed and direction, we collected the U10 and V10 variables, representing wind components at 10 meters, from the ERA5 reanalysis dataset archived on the European

Centre for Medium-Range Weather Forecasts (ECMWF) platform. Additionally, we obtained the precipitation data observed by the regional meteorological station in Putian (119.00°E, 25.44°N) for further analysis.

### 2.2 Regional Chemical Transport Model

WRF-Chem is one of the most widely used regional chemical transport models in atmospheric chemistry modeling (Zhang et al., 2024a). It can simulate the evolution of ambient trace gases and aerosols with fully coupled meteorology-

chemistry feedback (Grell et al., 2005). This study used WRF-Chem (version 4.1.5) to explore coastal HONO formation and the associated environmental influences. To align with the measurement campaign, the model was run from 26 April to 31 May in 2024. The first five days were reserved for spin-up and were excluded from the data analysis. The entire simulation was conducted in a seven-day loop cycle to avoid systemic biases. The modeling domain focused on the southeastern coastal region of China (Figure 1), with a grid spacing of 10 km. Fujian Province is highlighted in a blue

dashed box in Figure 1a. The coastal regions of Fujian consist of plains with a terrain elevation of less than 500 meter. Central Fujian is occupied by longitudinal mountains. This study focuses on six coastal cities for further discussion. As shown in Figure 2b, these cities are, from north to south: Ningde (ND), Fuzhou (FZ), Putian (PT), Quanzhou (QZ), Xiamen (XM), and Zhangzhou (ZZ). The observational site DZSK is located in the suburbs of Putian. In addition, 30 vertical layers were set up, extending from the ground surface to a height of 50 hPa.

The initial and lateral meteorological input conditions were derived from the final (FNL) global reanalysis dataset of the National Oceanic and Atmospheric Administration (NOAA). We also utilized four-dimensional data assimilation (FDDA)





to constrain gridded surface and upper atmospheric parameters with extra weather observational datasets. Continental anthropogenic emissions were obtained from the Multi-resolution Emission Inventory for China (MEIC version 1.4) (Li et al., 2017; Zheng et al., 2018). Shipping emissions were obtained from the Shipping Emission Inventory Model (SEIM),

which was developed by Tsinghua University (Liu et al., 2016). Figure S1 illustrates the spatial distribution pattern of continental and oceanic shipping emissions within the modeling domain. Biogenic emissions were calculated online using the Model of Emissions of Gases and Aerosols from Nature (MEGAN, version 2.0.6) (Guenther et al., 2006). Furthermore, this study adopted various parameterizations to predict sub-grid atmospheric physical and chemical processes. The detailed information was compiled in Table 1. The Statewide Air Pollution Research Center (SAPRC99)

mechanism (Carter, 2000) and the Model for Simulating Aerosol Interactions and Chemistry (MOSAIC) (Zaveri et al., 2008) were selected for gas-phase and aerosol chemistry simulations, respectively. The Fast-J module was chosen to calculate the photolytic rates of trace gases (Wild et al., 2000).

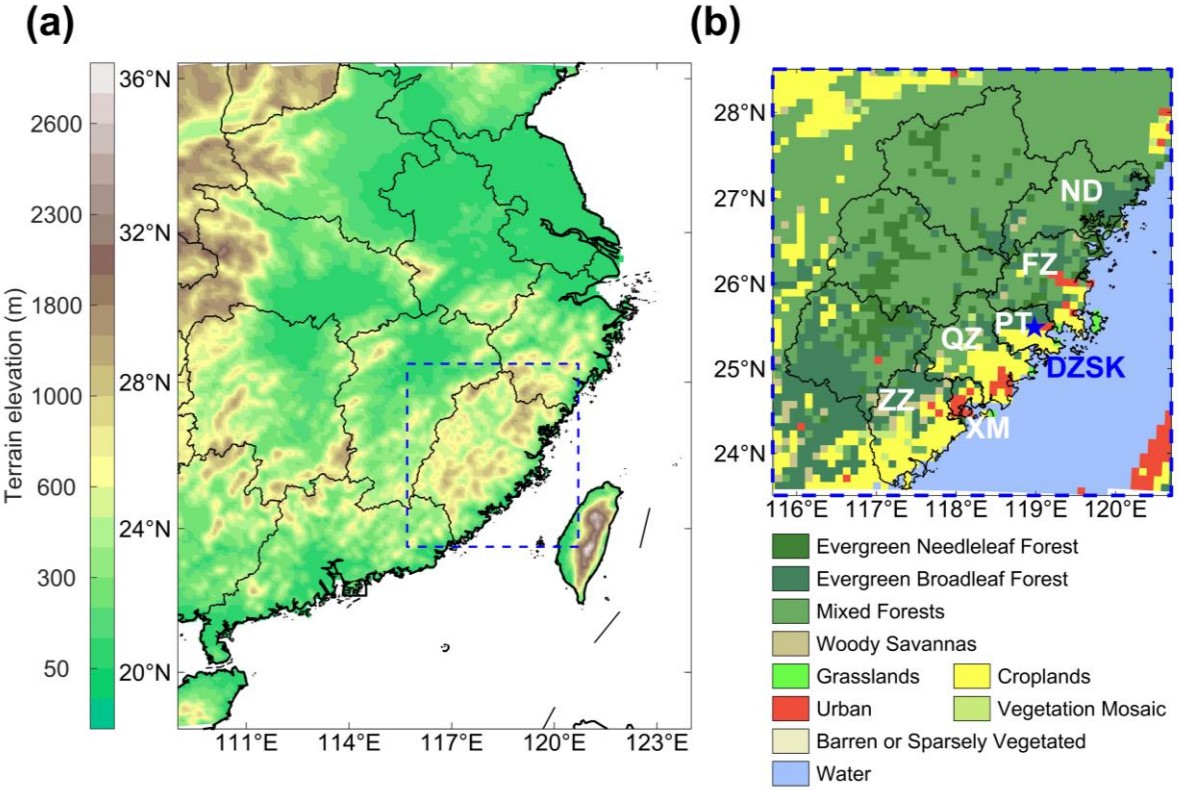

**Figure 1.** Design of the WRF-Chem modeling domain used in this study. Panel (a) depicts the terrain elevation of the
modeling domain. Panel (b) illustrates the land use information in Fujian Province. The DZSK site (blue pentagram) and
the six coastal cities (white bold font) are also marked in Panel (b).



**Table 1.** List of WRF-Chem modeling configurations.

|  | Parameter | Configuration |
|---|---|---|
| Time setup | Simulation duration | From 26[th] Apr to 31[st] May, 2024 |
|  | Spin-up period | 5 days |
| Domain setup | Grid number | 180 × 210 |
|  | Domain center | 116.2 °E, 27.8 °N |
|  | Horizontal resolution | 10 km × 10 km |
|  | Vertical configuration | 30 layers from surface to 50 hPa height |
|  | Projection | Lambert conformal |
| Physical parameterization | Microphysics | Lin (Lin et al., 1983) |
|  | Long-wave radiation | RRTMG (Iacono et al., 2008) |
|  | Short-wave radiation | RRTMG (Iacono et al., 2008) |
|  | Surface layer | MM5 Monin-Obukhov (Jiménez et al., 2012) |
|  | Land surface | Noah (Chen and Dudhia, 2001) |
|  | Boundary layer | YSU (Hong et al., 2006) |
|  | Cumulus | Grell 3D (Grell and Dévényi, 2002) |
| Chemical parameterization | Gas-phase reactions | SAPRC99 (Carter, 2000) |
|  | Aerosol processes | MOSAIC (Zaveri et al., 2008) |
|  | Photolysis | Fast-J (Wild et al., 2000) |

## 2.3 HONO Source Updates and Process Analyses

In the original WRF-Chem model, the default source of HONO formation is the homogeneous reaction between NO and

OH radicals. In a previous study, we incorporated multiple additional HONO sources into the WRF-Chem model (Zhang et al., 2024b), including direct emissions from fuel combustion processes, photo- and dark-oxidation of $NO_x$, light-enhanced heterogeneous uptake of $NO_2$ on the ground and aerosol surfaces, and photolysis of particulate nitrate aerosols. Updates to the chemical parameterizations of the HONO sources are summarized in Table 2. The heterogeneous uptake of $NO_2$ by ground and aerosol surfaces was light-dependent. We assumed that the nighttime uptake coefficients ($\gamma$) on the

ground and aerosol surfaces were $8\times10^{-6}$ and $4\times10^{-6}$, respectively (Zhang et al., 2021). During the daytime, the enhanced sunlight condition could increase the $\gamma$ value to a maximum of $1\times10^{-3}$ for the aerosol surface and $6\times10^{-5}$ for the ground surface. For the remaining daylight hours, we applied a dynamic linear scaling formula adjusted by solar radiation intensity, as presented by Song et al. (2023). In addition, two key parameters have been updated in this study to more accurately represent the HONO formation rate in typical coastal regions. The first parameter is the ratio of HONO to $NO_x$

($HONO/NO_x$), which reflects emission intensity. Here, we utilized the value of 1.45% proposed by Hu et al. (2022) based on long-term in situ measurements in Xiamen, Fujian. This value more accurately represents direct HONO emissions from local fuel combustion. Simultaneously, we estimated HONO emissions from the shipping sector using the same ratio. Secondly, we set the photolysis frequency of nitrate aerosols ($J_{no3-}$) to 120 times that of gaseous $HNO_3$ (Fu et al., 2019; Ye et al., 2016, 2017).



**Table 2.** Chemical parameterizations of HONO sources in the revised WRF-Chem model.

| Pathway | Parametrization | Descriptions |
|---|---|---|
| Direct Emissions | $E_{HONO} = \dfrac{HONO}{NO_x} \times E_{NO_x}$ | $E_{HONO}$ and $E_{NO_x}$ are emissions of HONO and $NO_x$. $HONO/NO_x = 1.45\%$ (Hu et al., 2022) |
| Heterogeneous uptake of $NO_2$ by ground surface | $k_g = \dfrac{1}{8} \times v_{NO2} \times \dfrac{S}{V} \times \gamma$ | $k_g$, $V_{NO2}$, $S/V$, and $\gamma$ are uptake reaction rate of $NO_2$ by the ground surface ($s^{-1}$), average molecular velocity (cm $s^{-1}$), ground surface area density ($cm^2$ $cm^{-3}$), and the unitless uptake coefficient (Zhang et al., 2024b). |
| Heterogeneous uptake of $NO_2$ by aerosol surface | $k_a = \dfrac{1}{4} \times v_{NO2} \times \dfrac{S}{V} \times \gamma$ | $k_a$, $V_{NO2}$, $S/V$, and $\gamma$ are uptake reaction rate of $NO_2$ by the aerosol surface ($s^{-1}$), average molecular velocity (cm $s^{-1}$), aerosol surface area density ($cm^2$ $cm^{-3}$), and the unitless uptake coefficient (Zhang et al., 2024b). |
| Photolysis of nitrate aerosols | $J_{NO3^-} = 120 \times J_{HNO3}$ | $J_{NO3^-}$ and $J_{HNO3}$ are the photolysis frequencies of nitrate aerosols and gaseous $HNO_3$ (Fu et al., 2019). |
| Photo- and dark-oxidation of $NO_x$ | $HNO_3 + NO \rightarrow NO_2 + HONO$ <br> $NO_3 + NO \rightarrow 1.98\ NO_2 + 0.02\ HONO$ | Laboratory-based reaction kinetics from Song et al. (2023). |

We used a source-oriented method (SOM) to determine the proportion of HONO production and consumption processes contributed by each source. A detailed description of the SOM diagnostic module can be found in Zhang et al. (2024b). In the SOM analysis, seven variables were configured to trace the formation process of HONO. These variables included the homogeneous reaction between NO and OH (NO+OH), $NO_x$ photo-oxidation, $NO_x$ dark-oxidation, primary emissions, the heterogeneous uptake of $NO_2$ by aerosols (Hetero-aerosol), the heterogeneous uptake of $NO_2$ on the ground surface (Hetero-land) and the photolysis of nitrate aerosols (Nitrate-photolysis). The first three sources are categorized as gaseous reaction, while the latter three are categorized as surface reactions. We also quantified two HONO chemical sink pathways: photodissociation of HONO (HONO+hv) as well as OH-oxidation removal (HONO+OH). Similarly, we also set up the SOM analysis to investigate the formation of OH radicals. Five formation pathways were identified, including the gas-phase reaction between $HO_2$ and NO ($HO_2$+OH), the reaction between atomic oxygen ($O^1D$) and $H_2O$ ($O^1D$+$H_2O$) initiated by $O_3$ photolysis ($O_3$+hv → $O^1D$+$O_2$), photolysis of HONO (HONO+hv), photolysis of hydrogen peroxide ($H_2O_2$+hv), and ozonolysis of VOCs ($O_3$+VOCs). All of these chemical reactions are primary sources of OH radical formation except $HO_2$+OH, which is generally considered as a secondary conversion (Xue et al., 2025).



**2.4 Sensitivity Experiment Designs**

Our study also conducted sensitivity experiments to quantify the contribution of oceanic shipping emissions to the formation of HONO in coastal areas, and the contribution of HONO chemistry to the enhancement of ambient OH radicals and $O_3$. In total, three simulation cases were run, using the same meteorological initial and lateral conditions and continental anthropogenic and biogenic emissions. As listed in Table S1, the first case was the BASE case, which considered oceanic shipping emissions but did not include the updated HONO sources. The REV case included both

oceanic shipping emissions and the updated HONO formation pathways. The "Noship" case denotes a simulation that used an improved HONO source representation but excluded oceanic shipping emissions. By comparing the BASE case (or Noship) with the REV case, we estimated the respective contribution of the updated chemical sources and shipping emissions to HONO formation.

**3 Results and Discussions**

**3.1 Overview of Field Observations**

Figure 2 shows hourly observations of air pollutants and meteorological parameters at the DZSK site from 1 to 31 May 2024. Generally, HONO concentrations ranged from 0.01 to 1.11 ppbv, with an average of 0.23 ppbv. We compared the HONO measurements at the DZSK site with in situ observations collected from previous studies in China. Table S2 presents a dataset of HONO measurements from 43 studies conducted between 2008 and 2025. It can be concluded that

observational studies were intensively carried out in urban areas of the NCP region, particularly in Beijing, where recorded HONO concentrations frequently exceeded 1 ppbv. These higher HONO levels were attributed to abundant $NO_x$ emissions. By contrast, HONO concentrations were found significantly lower in remote regions. For instance, HONO concentrations ranging from 0.13 to 0.15 ppbv were measured at the summit of Mt. Tai (Jiang et al., 2020; Xue et al., 2022). Studies of coastal regions witnessed HONO concentrations of around 0.50 ppbv (Hu et al., 2022; Zhong et al.,

2023). Overall, the HONO levels measured at the DZSK site were much lower than those reported by previous measurements in China. The mean $NO_2$ concentration during the study period was 6.3 μg m$^{-3}$, due to low $NO_x$ emissions in the suburban, which probably contributed to the low HONO concentrations. Meanwhile, the average wind speed was 2.1 m s$^{-1}$, providing a favourable diffusion condition for air pollutants. The prevailing wind direction at the DZSK site was from the northeast, implying a potential contribution from oceanic shipping emissions.

Figure S2 exhibits the diurnal variations of gaseous air pollutants and meteorological parameters. On non-rainy days, air temperature and solar radiation were higher, with an apparent peak occurring around midday. Substantial sunlight



accelerated the photochemical production of air pollutants such as O₃. Consequently, concentrations of air pollutants were obviously higher on non-rainy days than on rainy days. At the same time, precipitation also led to the wet deposition of air pollutants. To avoid the influence of precipitation, we focused on the diurnal variation pattern of HONO

concentrations excluding rainy days. The maximum hourly HONO concentration at the DZSK site occurred at 14:00 on non-rainy days, peaking at 0.38 ppbv. This phenomenon is contrary to the diurnal variations of HONO observed in previous inland measurements. According to Wang et al. (2025), HONO concentrations in inland urban and rural areas typically displayed the minimum in the afternoon. Despite the decrease in air temperature and radiation after 14:00, the O₃ concentration increased until it reached a daily maximum of 125 μg m$^{-3}$ at 16:00. To explore the mechanisms of

HONO formation in coastal regions, especially the diurnal peak during the daytime, our study conducted SOM process analyses in the subsequent sections. The potential contribution of shipping emissions to coastal HONO formation was also quantified using sensitivity experiments.

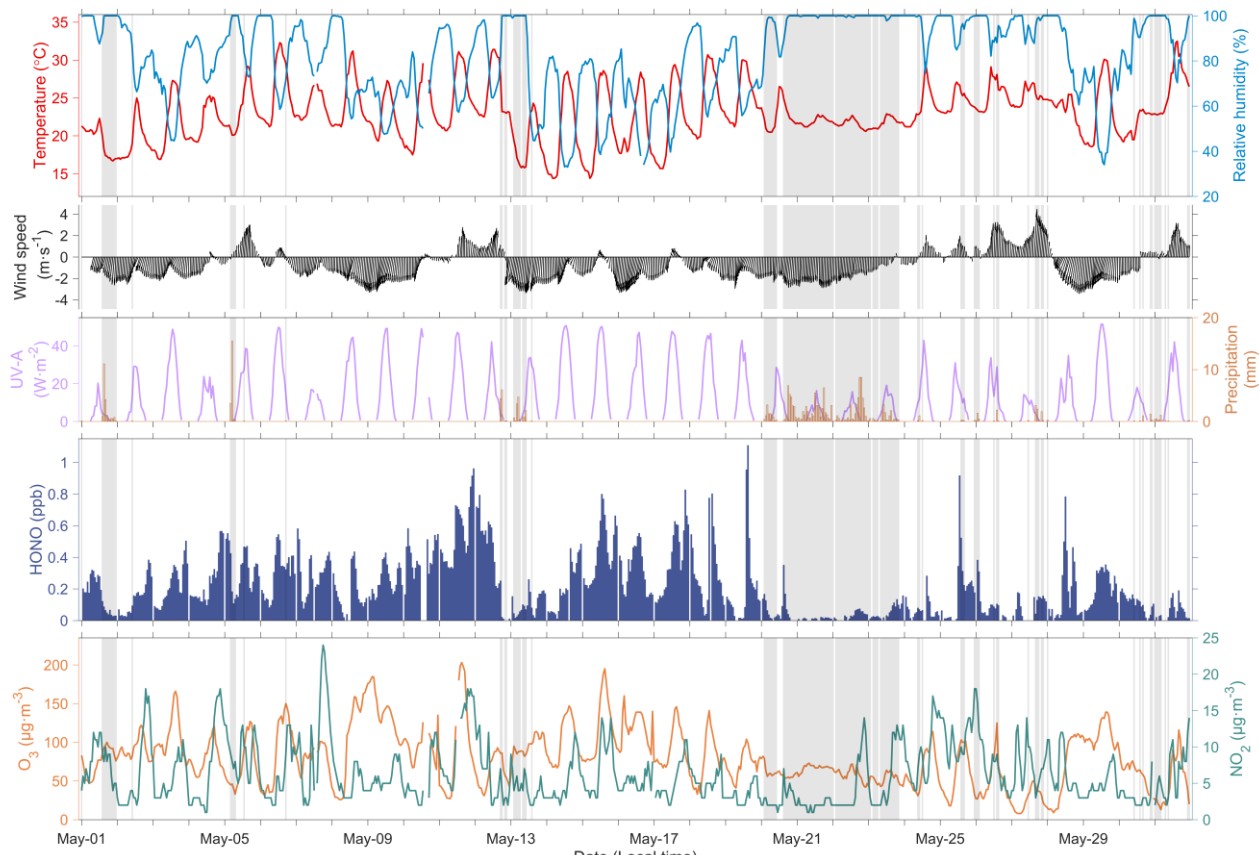

**Figure 2.** Hourly measurements of meteorological parameters and gaseous air pollutants at the DZSK site during 1–31

May 2024. The shaded areas stand for the period of precipitation.





### 3.2 Evaluations of the Numerical Model

First, we conducted a model evaluation based on measurements at the DZSK site. Figure 3 demonstrates a comparison between the simulated and observed HONO concentrations. The BASE simulation failed to capture the magnitude and temporal variations of HONO concentrations at the DZSK site. Including multiple HONO formation pathways greatly

improved the model performance of HONO (Figure 3a). The monthly mean HONO concentration increased from 0.03 ppbv (BASE) to 0.25 ppbv (REV). We then applied three statistical metrics to reveal improvements in HONO modeling. The calculation of validation indicators has been elucidated in our previous modeling studies (Zhang et al., 2024b). As summarized in Table 3, the index of agreement (IOA, varies from 0 to 1) for predicting hourly HONO concentrations increased from 0.62 (BASE) to 0.69 (REV). Simultaneously, the normalized mean bias (NMB, varies from $-\infty$ to $+\infty$)

and the root mean square error (RMSE, varies from 0 to $+\infty$) considerably decreased by 91% and 21%, respectively. While the revised model reasonably reproduced the observed temporal variations in HONO concentrations during the study period, an underestimation existed on 16–18 May, suggesting a potential omission of HONO sources. Additionally, the WRF-Chem model overestimated low HONO concentrations during 21–24 May, when there was continuous precipitation (Figure 2). Simulations including the updated HONO sources more accurately represented the diurnal

variation pattern of HONO concentrations. Figure 3c illustrates that the REV case successfully captured the higher HONO concentrations observed around noon. The Pearson's correlation coefficient (R) between the measurements and simulations increased from 0.657 (BASE) to 0.763 (REV).

**Table 3.** Statistical metrics of evaluating HONO simulations.

|  | Mean observation (ppbv) | Mean simulation (ppbv) | IOA | NMB | RMSE (ppbv) |
|---|---|---|---|---|---|
| BASE | 0.23 | 0.03 | 0.62 | − 86% | 0.28 |
| REV | 0.23 | 0.25 | 0.69 | + 8% | 0.22 |

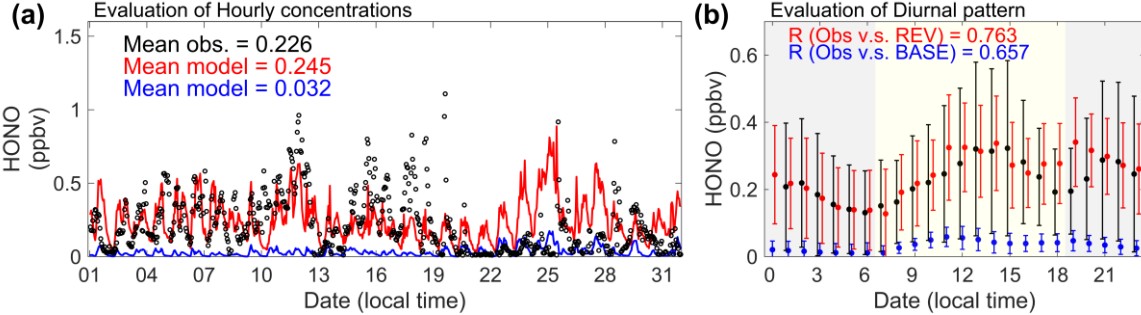

**Figure 3.** Model evaluations of HONO simulations at the DZSK site in May 2024. Panels (a) and (b) show comparisons between observations and simulations of hourly and diurnal HONO concentrations, respectively. The red and blue fonts represent the REV and BASE modeling cases, respectively. The lower and upper limits in panel (b) indicate the mean ± standard deviation (σ).





The model performance for important precursors and products of HONO chemistry, $NO_2$ and $O_3$, were improved as well.

As illustrated in Figure 4a and 4c, the diurnal variations in the simulated $NO_2$ and $O_3$ concentrations by REV were more consistent with the observed values. The diurnal Pearson's correlation coefficients between measurements and simulations for $NO_2$ and $O_3$ increased by 5% and 3%, respectively. At the same time, underestimation of $O_3$ concentrations in the original model was effectively reduced. The daily maximum $O_3$ concentration in the revised version (REV) was much closer to the observed value than in the BASE simulation. The estimated maximum daily average 8-

hour (MDA-8) concentration of $O_3$ increased from 62.9 $\mu g \ m^{-3}$ (BASE) to 85.6 $\mu g \ m^{-3}$ (REV), with an observed value of 103.0 $\mu g \ m^{-3}$ (Figure 4d). Accurately characterizing the HONO chemical budget can therefore improve the simulation of $O_3$ concentrations in 3D chemical transport models.

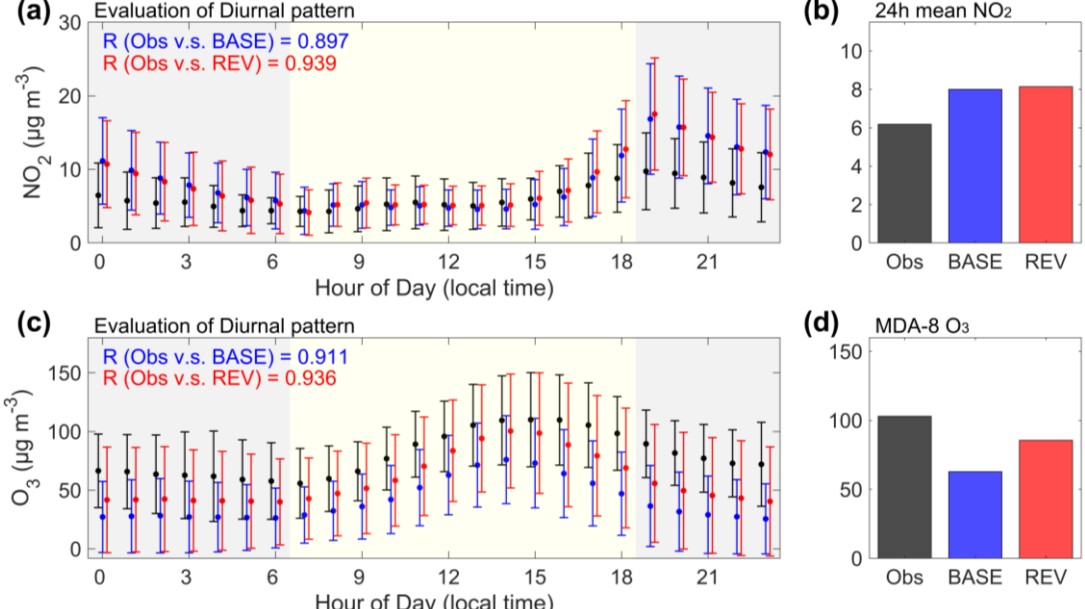

**Figure 4.** The same as Figure 3b, but for diurnal variations of $NO_2$ and $O_3$ concentrations.

**3.3 Analyses of HONO Formation Mechanisms**

**3.3.1 Spatio-temporal Characteristics**

This study used SOM process analysis to elucidate the mechanisms of HONO formation in Fujian's coastal regions. The analysis focused on the regional average from six coastal cities, as depicted in Figure 1. The diurnal variations in the chemical production and consumption rates of HONO are presented in Figure 5a. The maximum HONO production rate

of over 1.30 ppbv $h^{-1}$ occurred during the midday hours (11:00 to 14:00), due to the intensified formation pathways. The



HONO consumption rates were primarily influenced by self-photolysis (HONO+hv) and exhibited a diurnal pattern similar to that of the production rates. The average HONO production rate during the daytime (7:00 to 18:00) was found to be 0.97 ppbv h$^{-1}$ (Figure 5b). The SOM analysis revealed that gas-phase and surface reactions contributed equally to daytime HONO formation, accounting for 45% each. Specifically, the heterogeneous uptake of $NO_2$ on the ground

surface and photo-oxidation of $NO_x$ emerged as the two principal contributors to HONO formation in coastal regions, with an aggregate contribution of 64%. In descending order, other sources contributed 16% (NO+OH), 10% (primary emissions), 8% (nitrate-photolysis), 2% (hetero-aerosol), and less than 1% ($NO_x$ dark-oxidation) to daytime HONO formation. Although the heterogeneous uptake of $NO_2$ on the ground surface was identified as the predominant pathway, its relative importance evidently declined from morning to afternoon (Figure 5a). This decrease could be attributed to the

growing influence of nitrate photolysis, NO+OH, and $NO_x$ photo-oxidation. The mean nocturnal HONO production rate dropped to 0.12 ppbv h$^{-1}$, reflecting a substantial decline in the intensity of oxidation-related HONO formation pathways (Figure 5b). At the same time, HONO consumption rates approached zero at night due to the cessation of photochemical loss mechanisms. Overall, the heterogeneous $NO_2$ uptake on the ground surface remained the dominant source for nighttime HONO production, accounting for 48%. In the meantime, the contribution of direct emissions to nocturnal

HONO production significantly increased to 17%.

The mechanisms of HONO formation also proposed spatial heterogeneity. Our study examined HONO formation pathways across different land surfaces, such as forests, grasslands, farmlands, and urban areas (see Table S3). Figure 1b illustrates the spatial distribution of land use information within the study region. In forested areas, SOM analysis indicated that heterogeneous uptake of $NO_2$ by the ground surface accounted for 45% of HONO production. This

phenomenon is likely linked to the higher density of reactive surface area resulting from the larger leaf area index (LAI) of forested areas (Zhang et al., 2016). In urban areas, however, high $NO_x$ levels could facilitate greater HONO production through gas-phase reactions involving $NO_x$, particularly $NO_x$ photo-oxidation and NO+OH, which contributed 36% and 21%, respectively. The absolute HONO production rates from these two $NO_x$-related reactions in urban areas were approximately four times higher than in forests. Direct emissions also contributed significantly to HONO formation (23%)

due to intensive $NO_x$ emissions in urban areas. Since two-thirds of the coastal areas are covered by forests, the regional average contribution of $NO_2$ heterogeneous uptake on the ground surface exceeded that of any $NO_x$-related gas-phase oxidation reaction or direct emissions.

Previous investigations have indicated that light-enhanced heterogeneous uptake of $NO_2$ on the ground surface is the dominant source of HONO formation in inland regions (Zhang et al., 2024b). Our diagnostic results in the coastal regions

of Fujian support this finding. However, the specific contribution in Fujian (35–48%) was obviously lower than that observed in inland areas (42–86%). Similarly, the contribution from the heterogeneous $NO_2$ uptake on aerosol surfaces





(1–2%) was lower than that reported for inland areas (3–20%), because of lower particle concentrations in coastal regions. Moreover, the present study pointed out a greater contribution to HONO formation from $NO_x$-related gas-phase oxidation reactions than was found in previous studies.

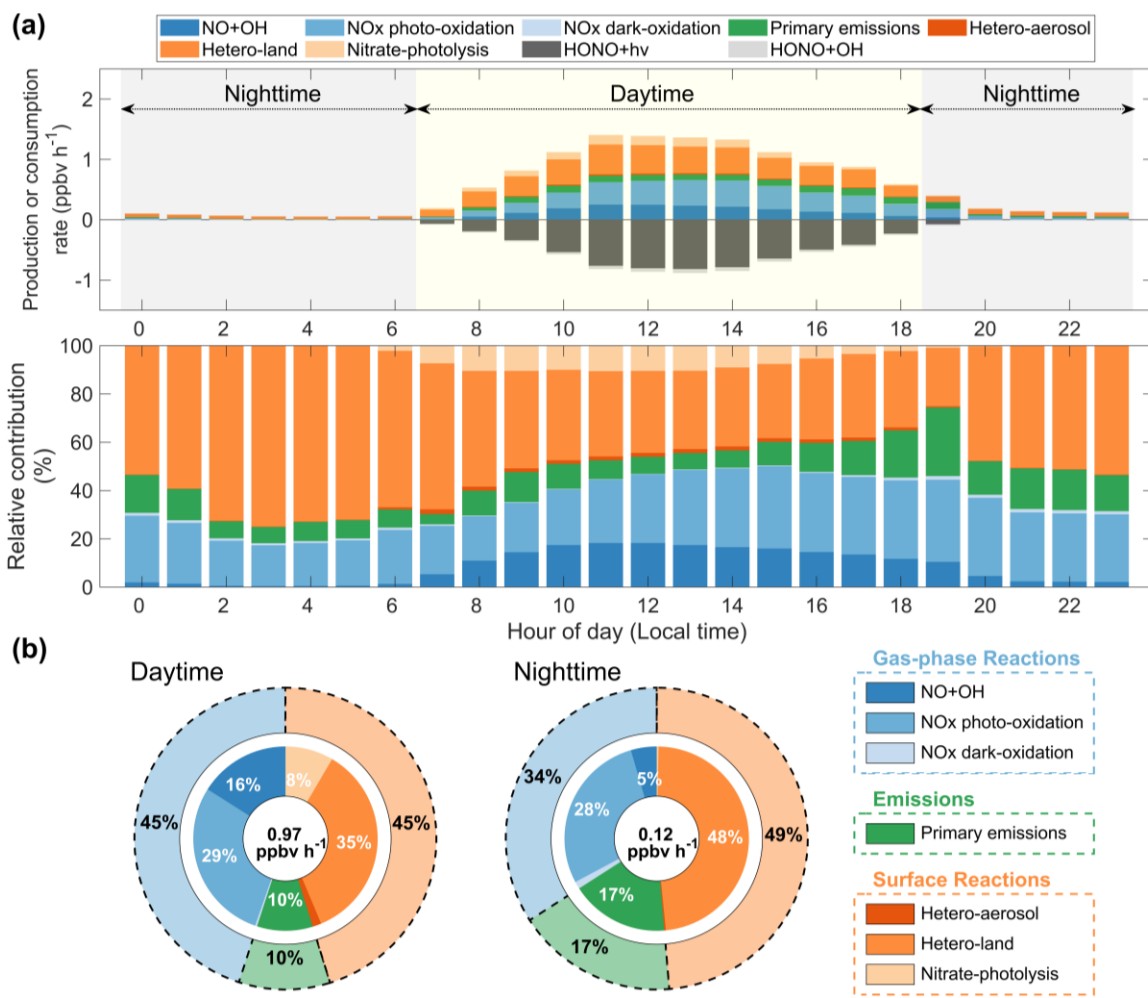


**Figure 5.** Attribution of mechanisms of HONO formation in coastal Fujian in May 2024. Panel (a) shows the diurnal variations in the absolute and relative contributions of seven production and two consumption reactions to the HONO chemical budget. Panel (b) illustrates the contribution of the seven production reactions to daytime and nighttime HONO formation during the study period. The total production rates are labelled in the middle of the pie charts.

**3.3.2 Impact of Shipping Emissions**

Furthermore, we conducted sensitivity experiments to evaluate the potential impact of shipping emissions on HONO formation in coastal regions. Figure 6 demonstrates the spatial distribution of $NO_x$, $NO_3^-$, and HONO concentrations in





coastal regions that can be attributed to shipping emissions. As $NO_x$ is the most important precursor for HONO formation, we first examined the influence of shipping emissions on coastal $NO_x$ concentrations. The results indicated that shipping

emissions caused a net increase of 0.68 ppbv in $NO_x$ levels across the coastal regions of Fujian, with a clear decrease in gradient from the ocean towards inland areas. It is revealed that shipping emissions contributed 17% to coastal $NO_x$ concentrations, with a higher contribution in the northern regions than in the middle urban areas. The subsequent rise in $NO_x$ concentrations led to an increase in nitrate aerosol concentrations, which probably facilitated HONO formation. Spatially, shipping emissions induced a net elevation in the average $NO_3^-$ concentration of 0.52 µg m$^{-3}$, accounting for

approximately 33% of the total. The spatial distribution of nitrate aerosol concentrations differed from that of $NO_x$, with higher levels concentrated mainly in areas with intensive $NH_3$ emissions. Consequently, HONO levels in the coastal regions of Fujian increased by 36 pptv due to increased precursor concentrations resulting from shipping emissions, representing an 18% relative contribution. Similar to the $NO_x$ spatial distribution pattern, shipping emissions promoted higher HONO levels in the northern areas of the study region.

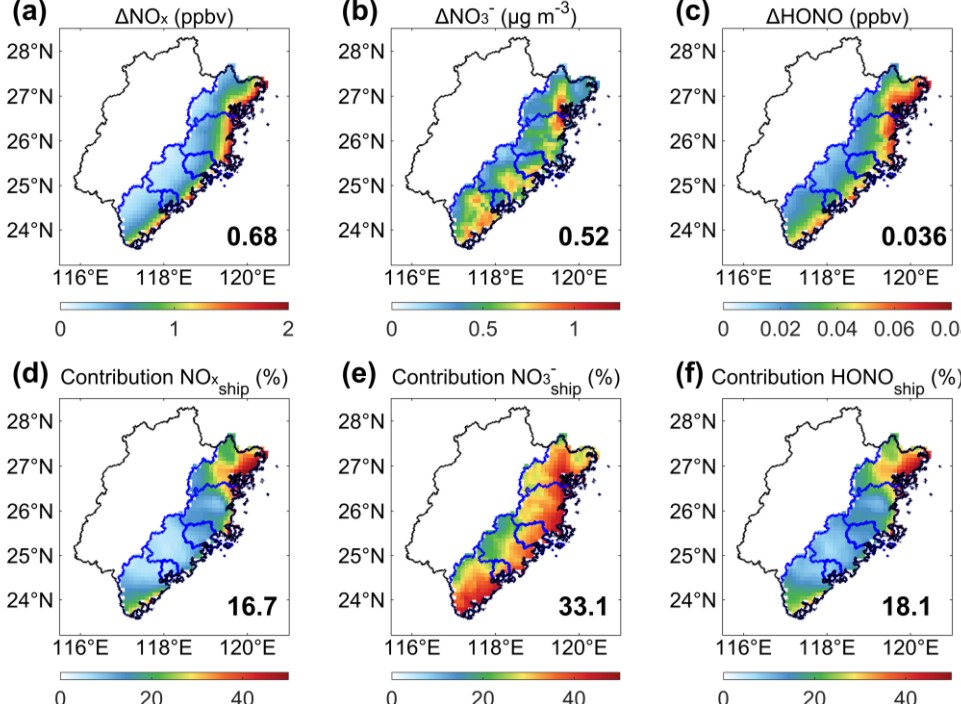

**Figure 6.** The simulated regional mean effect of shipping emissions on the concentrations of HONO and its precursors, including $NO_x$ and $NO_3^-$ aerosols, during the study period. Panels (a-c) show the absolute changes, while panels (d-f) represent the relative contributions from shipping emissions ($\frac{REV-Noship}{REV} \times 100\%$). The regional average values are marked in the bottom right-hand corner of each panel.





As illustrated in Figure S3, the simulated diurnal range of coastal NO$_x$ concentrations attributed to shipping emissions was from 0.43 to 0.99 ppbv. This effect was more pronounced during the nighttime hours. Specifically, NO$_x$ released from the shipping sector preferred to react with O$_3$ near the ocean surface. As depicted in Figure S4, the chemical production rates of O$_3$ during both daytime and nighttime were negative over coastal waters with substantial shipping emissions. This suggests that the transport of NO$_x$ to coastal regions was probably inhibited. Meanwhile, the O$_3$ titration

effect was more evident during the daytime than at night due to higher shipping emissions, meaning that shipping emissions contributed less to coastal NO$_x$ during the daytime. Therefore, nitrate aerosols, the products of NO$_x$ oxidation, exhibited the same diurnal variation pattern as NO$_x$ with respect to contributions induced by shipping emissions (Figure S3). Despite the fact that NO$_x$ and nitrate aerosols contributed by shipping emissions were relatively lower during the daytime, HONO production rates were higher, particularly around noon (Figure 7). This phenomenon could be attributed

to light-dependent reactive pathways that efficiently increased HONO production rates in the presence of sunlight. The mean increased production rate of HONO in coastal regions resulting from shipping emissions during the daytime was 0.15 ppbv h$^{-1}$. Heterogeneous uptake of NO$_2$ on the ground surface, NO$_x$ photo-oxidation, NO+OH, and nitrate photolysis accounted for 39%, 34%, 13%, and 12% of the total enhancement, respectively. In contrast, HONO production rates attributed to shipping emissions were much lower in the evening. However, due to the self-photolysis of HONO, the

overall increase in coastal HONO concentrations caused by shipping emissions during the daytime (0.19 ppbv) was close to daytime levels (0.20 ppbv).

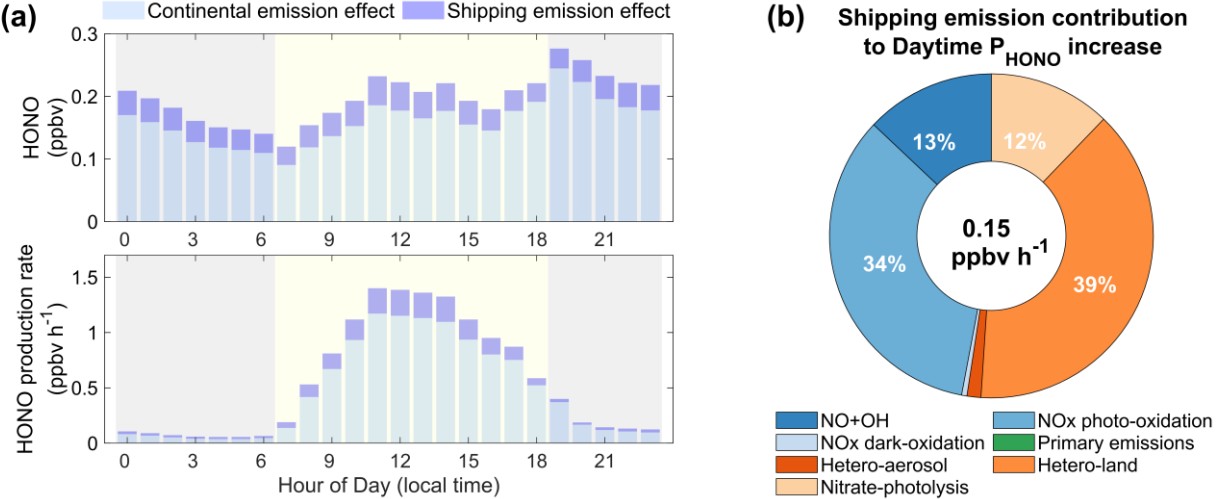

**Figure 7.** The simulated shipping emission contributions to concentrations and formation rates of HONO over the study region. Panel (a) shows the diurnal variations in the effects of continental and shipping emissions on HONO

concentrations and formation rates. Panel (b) displays the contribution of shipping emissions to the daytime HONO production rate from different sources.



### 3.3.3 Attribution of Daytime High HONO Concentrations

We subsequently interpreted the causes of the high HONO levels in the coastal regions of Fujian. The predicted high HONO concentrations over the study region between 11:00 and 14:00 were attributed to a simultaneous increase in

chemical production rates (see Figures 5a and 7a). The rapid conversion efficiency offset the effect of low $NO_x$ and nitrate aerosol concentrations, leading to higher HONO production rates around noon. Meanwhile, the aggregate HONO production rate exceeded consumption rates resulting from self-photolysis and OH-oxidation reactions. This contributed to the accumulation of high HONO concentrations. Previous studies concluded that the daytime peak in HONO concentrations was possibly driven by nitrate photolysis (Hu et al., 2022; Wang et al., 2025). However, our attribution

analysis inferred that nitrate photolysis was not the dominant factor. Even when an upper limit of the empirical photolysis frequency of nitrate aerosols ($120J_{HNO3}$) was applied (Fu et al., 2019; Zhang et al., 2022a), the low nitrate concentration of about 1.0 μg m$^{-3}$ only accounted for less than 10% of the overall production rate during the daytime (Figure S3). This study suggests that heterogeneous $NO_2$ uptake on the ground surface and $NO_x$ photo-oxidation were the main drivers of HONO formation around noon (11:00 to 14:00), contributing 30% and 34%, respectively. The high

production rates of these two photo-chemical reactions could be attributed to the persistent abundant solar radiation in coastal areas. Analysis of the spatial distribution patterns of HONO formation mechanisms revealed that the $NO_2$ heterogeneous uptake reaction on the ground surface was more effective in forests, whereas the photo-oxidation of $NO_x$ played a more important role in coastal urban areas (Table S3). In addition, HONO photolysis could release OH radicals, facilitating further photo-oxidation of $NO_x$, particularly at noon when the intensity of photochemistry reaches its diurnal

peak. We also quantified the contribution of shipping emissions to coastal HONO levels in Figure 7. We found that 0.22 ppbv h$^{-1}$ of the HONO production rate was attributable to shipping emissions around noon, resulting in a 20% increase in HONO concentrations. These results emphasize the importance of including shipping emissions in the HONO budget near coastal regions.

### 3.4 Contributions of HONO to Ambient Oxidants

**3.4.1 Enhancement of OH Radicals**

We assessed the role of HONO in OH formation over the coastal regions of Fujian utilizing the SOM process analysis. As illustrated in Figure 8, the OH production rates from four primary sources and one secondary source are presented. The total OH production rate peaked at 14:00, reaching a maximum value of 7.78 ppbv h$^{-1}$ (Figure 8a). Conversely, OH production rates at night were significantly lower, remaining below 1 ppbv h$^{-1}$. The average daily OH production rate

throughout the study period was calculated to be 2.61 ppbv h$^{-1}$. Notably, secondary conversion from the reaction between





$HO_2$ and NO was found to dominate OH production, contributing 68% of the total (Figure 8b). This fraction is consistent with the results of previous field measurement studies (Yang et al., 2021; Ye et al., 2023). Among primary sources identified, the reaction between $O^1D$ and $H_2O$ initiated by $O_3$ photolysis, and the photolysis of HONO emerged as two significant contributors to OH formation. These two pathways accounted for 15% and 10% of the daily OH production

rate. Evidently, $O^1D+H_2O$ and HONO+hv were more influential during daylight hours, with the combined contribution to the OH production rate peaking at 32% at 13:00.

Figure 8c illustrates the diurnal variations in the relative contributions to OH formation from four primary sources. Model results indicated that the contribution of HONO photolysis to OH formation was notably higher during morning hours. As $O_3$ concentrations increased, the reaction between $O^1D$ and $H_2O$ replaced HONO photolysis as the dominant

source for OH formation. This phenomenon has been confirmed by previous observational studies (Hu et al., 2022; Xue et al., 2025). The $O^1D+H_2O$ reaction reached a maximum contribution of approximately 60% to OH formation at 14:00. After this peak, the production rates of all reactions decreased due to diminishing sunlight conditions. Generally, the average daytime production rate of OH in the coastal regions of Fujian was estimated to be 1.52 ppbv $h^{-1}$. The contributions of the $O^1D+H_2O$, HONO+hv, photolysis of $H_2O_2$, and ozonolysis of VOC the primary

OH production rate were 51%, 34%, 8%, and 8%, respectively.

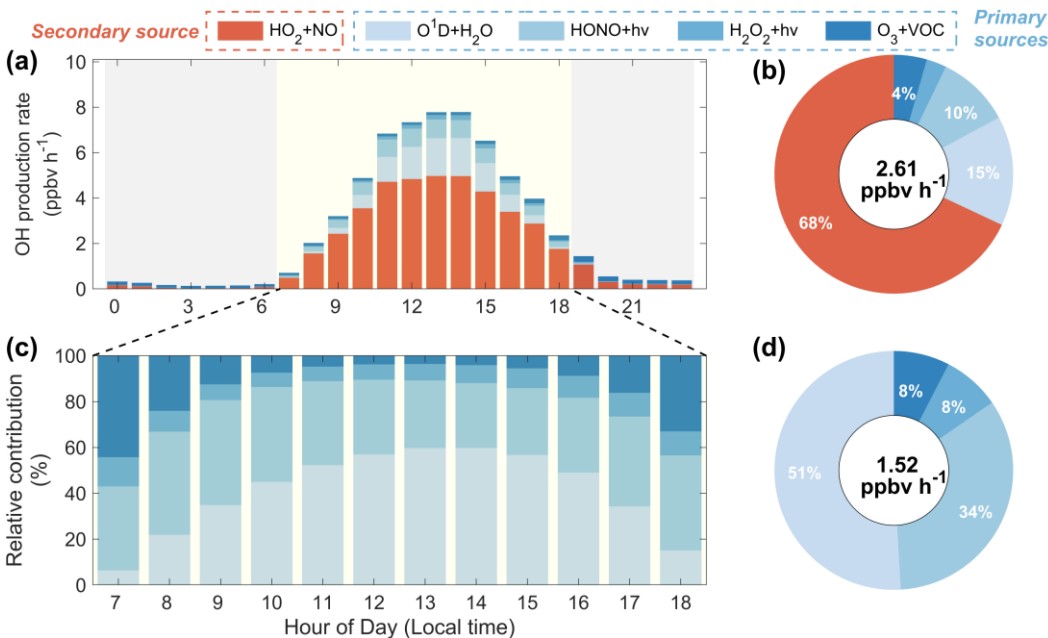

**Figure 8.** SOM process analysis of OH production rates in the coastal region of Fujian in May 2024. Panel (a) shows the diurnal variations of OH production rates from five sources. Panel (b) shows the relative contribution of each pathway. Panels (c-d) show the specific contribution of the four primary sources to OH formation during  the daytime.



Through sensitivity experiments, we further elucidated the impact of HONO on OH concentrations. As shown in Figure 9a, the diurnal OH concentrations in the coastal regions of Fujian experienced a remarkable increase due to the improved representation of HONO sources in the WRF-Chem model. Compared to the BASE simulation, the average daytime concentration of OH radicals increased by 57%. Concurrently, the daily maximum OH concentration rose from $7.5 \times 10^6$ molecules cm$^{-3}$ (BASE) to $12.1 \times 10^6$ molecules cm$^{-3}$ (REV). This increase in OH concentration can be attributed to the

alteration in OH production rate. The SOM process analysis indicated a 35% increase in the mean OH production rate.

By integrating sensitivity experiments with the SOM process analysis, we identified two principle factors that contributed to the enhancement of OH formation (Figure 9b). The first factor is the consequence of HONO photolysis, which increased the OH production rate by 0.40 ppbv h$^{-1}$. Due to the low HONO concentration in the BASE simulation, the contribution of the HONO photolysis to the OH production rate relatively surged by 378%. The second factor is the

promotion of ambient AOC conditions. The increased OH levels then enhanced the formation of oxidative products, including HO$_2$ radicals, H$_2$O$_2$, and O$_3$. The rise in O$_3$ further accelerated the generation of O$^1$D atom. Consequently, the chemical reactions involving HO$_2$+NO, O$^1$D+H$_2$O, and H$_2$O$_2$+hv led to increases in the OH production rate of 0.63, 0.18, and 0.04 ppbv h$^{-1}$, with relative changes of 23%, 29%, and 53%, respectively. The contribution to OH production from the O$_3$+VOC reaction was negligible as this reaction was more efficient at night and inactive during the daytime. In

conclusion, the direct increase in OH formation via HONO photolysis accounted for 32% of the overall effect attributed to HONO chemistry. The impact of the enhancement in the OH production rate contributed by the other four sources was twice that contributed by the photolysis of HONO (Figure 9c).

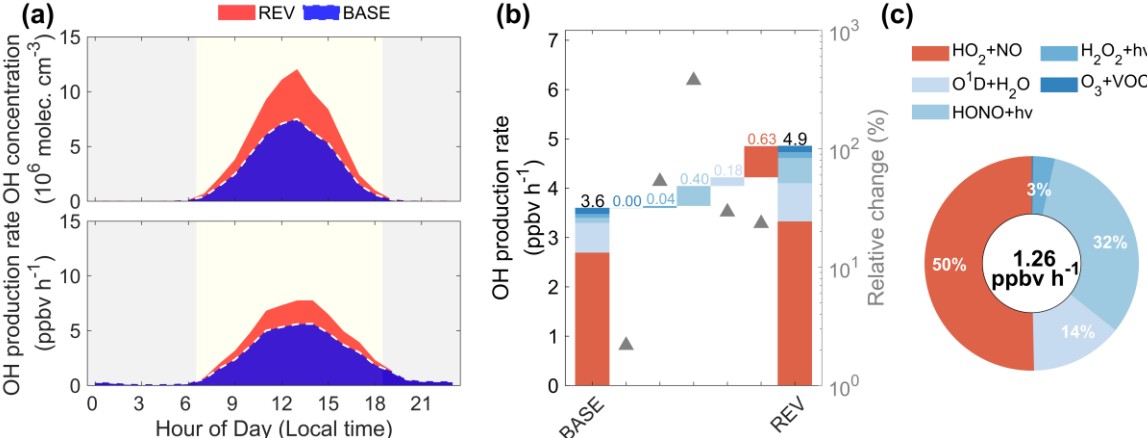

**Figure 9.** Comparisons of HONO concentrations and production rates in coastal Fujian in two simulation cases (BASE
and REV). Panel (a) displays the diurnal variations in OH concentrations and production rates. Panel (b) shows the absolute and relative changes in daytime OH formation rates from five chemical sources by comparing the BASE and REV cases. The relative contribution of these sources to the enhancement of the OH production rate is exhibited in Panel (c), with the number of increased production rate labelled in the middle of the pie chart.




### 3.4.2 Enhancement of O₃ Concentration

The sensitivity experiment simultaneously assessed the impact of HONO chemistry on $O_3$ concentrations. As discussed in Section 3.2, the model evaluation of $O_3$ levels has revealed that the discrepancy between the BASE and REV cases arises from increased AOC conditions due to the updates of HONO representation. Switching from the BASE simulation to the REV simulation led to an increase in 24-hour average $O_3$ concentration by 9.2 ppbv at the DZSK site, representing a 43% relative increase. In the coastal regions of Fujian, the regional mean absolute enhancement of $O_3$ concentration was 9.9

ppbv, corresponding to a relative increase of 44%, comparable to the findings at DZSK. Moreover, we compared the simulated absolute and relative alternations in $O_3$ concentrations attributable to HONO chemistry with the results of sensitivity experiments utilizing 3D chemical transport models obtained from previous studies. The related comparisons are summarized in Table 4. Most of the referenced studies referenced were conducted in densely populated areas in eastern China. The compilation suggests that the absolute increase in $O_3$ concentration driven by HONO chemistry across

most studies was approximately 10 ppbv and consistent with the value presented by this study. However, the relative enhancement of $O_3$ levels in this study (43–44%) was significantly higher than the range reported in previous studies (12–38%), exception for the outlier proposed by Fu et al. (2019). This discrepancy indicates a more pronounced catalytic impact of HONO on photochemical air pollution in coastal regions despite their comparatively lower concentrations of HONO.

**Table 4.** Compilation of contributions of HONO photolysis to enhancement of $O_3$ concentrations from previous studies.

| Period | Region | Model | Absolute ΔO₃ (ppbv) | Relative ΔO₃ (%) | References |
|---|---|---|---|---|---|
| 25–31 August 2011 | Hong Kong | WRF-Chem | 4.0 | 12 | (Zhang et al., 2016) |
| 26 June to 7 July 2014 | East China | WRF-Chem | 2.9–6.2 | 6–13 | (Zhang et al., 2017) |
| 4–8 January 2017 | Heshan, PRD | CMAQ | 24.0 | 70 | (Fu et al., 2019) |
| 22–31 July 2016 | NCP | | 13.0 | - | |
| 7–13 July 2013 | YRD | WRF-Chem | 14.7 | - | (Guo et al., 2020) |
| 18–31 October 2015 | PRD | | 10.7 | - | |
| 3 December 2015 to 14 January 2016 | Xi'an | WRF-Chem | - | 22 | (Li et al., 2022) |
| 1 March to 31 May 2016 | NCP | WRF-Chem | 7.9 | 17 | (Zhang et al., 2022b) |
| 1 July to 30 September 2016 | | | 8.9 | 16 | |
| 8–22 June 2017 | NCP | WRF-Chem | 26.1 | 38 | (Song et al., 2023) |
| 1–7 June 2017 | | | 6.9 | 16 | |
| 8–13 June 2017 | NCP | WRF-Chem | 9.5 | 20 | (Ran et al., 2024) |
| 14–21 June 2017 | | | 11.2 | 22 | |
| 24–27 March 2019 | Nanjing, YRD | WRF-Chem | 8.9 | 28 | (Zhang et al., 2024b) |
| 1–31 May 2024 | DZSK Coastal areas | WRF-Chem | 9.2 / 9.9 | 43 / 44 | This study |



### 3.5 Uncertainties

There are several uncertainties in HONO simulations that should be discussed. Firstly, the heterogeneous uptake coefficients of $NO_2$ by solid surfaces and the photolysis frequency of nitrate are highly uncertain. The maximum uptake coefficient $\gamma$ varies from $10^{-6}$ to $10^{-3}$ depending on sunlight conditions and surface properties (Zhang et al., 2024a).

Previous studies have widely used maximum $\gamma$ values of $1\times10^{-3}$ for aerosol surfaces and $6\times10^{-5}$ for the ground surface (Liu et al., 2019; Zhang et al., 2024b, 2021), demonstrating good robustness. Regarding the photolysis frequency, Zhang et al. (2022a) summarized that $J_{no3-}$ is 1–3 orders of magnitude higher than $J_{HNO3}$. Our study adopted the median magnitude ($120J_{HNO3}$) inferred from aircraft-based measurements conducted in the marine boundary layer over the North Atlantic ocean (Ye et al., 2016), as this better represents coastal areas.

Secondly, we used an empirical $HONO/NO_x$ value of 1.45% to estimate the direct HONO emissions from the fuel combustion process. This value was derived from long-term in situ measurements in coastal regions and proposed by Hu et al. (2022). We applied this value to describe fresh HONO emissions from ships. However, Ke et al. (2025) have pointed out that the same ratio of HONO to $NO_x$ may not be applicable to mobile sources, including mobile machinery, ships, and aircraft. Currently, there are no direct measurements of freshly emitted HONO from shipping. Therefore,

further observations and experiments are required to constrain HONO emissions from ships.

Thirdly, the ocean surface may affect HONO formation in other ways. Zha et al. (2014) observed significant evidence suggesting that the ocean was a nonnegligible contributor to coastal HONO production. Consequently, Zhang et al. (2016) parameterized this reaction as heterogeneous uptake by the ocean surface in the WRF-Chem model using a simplified formula. Additionally, an observational study proposed that seawater could increase the solubility of HONO under

certain conditions, indicating that the ocean surface could act as a sink for atmospheric HONO (Crilley et al., 2021; Wang et al., 2025). However, the source or sink effect of the ocean on HONO remains a controversial topic. Further studies on this theory and the relevant parameterization are needed in the future.

### 4 Conclusions

This study investigated the mechanisms of HONO chemical formation and its impact on the enhancement of ambient

oxidants in coastal Fujian of southeastern China. Based on continuous in situ measurements over a one-month period at a suburban site, we found that the observed HONO showed an unexpected diurnal variation pattern with higher levels measured at noon, contrary to an obvious daytime minimum reported by extensive previous studies focusing on inland areas. Using a revised WRF-Chem model incorporating updates to multiple sources of HONO, including direct emissions, $NO_2$ heterogeneous uptake by solid surfaces, nitrate photolysis, and the photo- and dark-oxidation of $NO_x$, we were able



440 to reasonably reproduce the magnitude and temporal variation of HONO concentrations, especially the higher values observed at noon.

To better understand the mechanisms of HONO formation in the coastal regions of Fujian, we subsequently utilized the SOM process analysis. In general, heterogeneous uptake of $NO_2$ on the ground surface and $NO_x$ photo-oxidation were two principal contributors to HONO formation, contributing 35–48% and 28–29%, respectively. From 11:00 to 14:00,

445 these two light-dependent reactive pathways led to rapid HONO production rates and offset the effect of lower precursor concentrations, accounting for 64% of the total. Notably, the formed HONO could release OH radicals by self-photolysis, thereby facilitating $NO_x$ photo-oxidation. In spatial, model results suggested that the heterogeneous uptake of $NO_2$ on the ground surface was more important in forest areas due to the higher density of reactive surface area. In urban areas, high $NO_x$ levels resulted in more HONO production through gas-phase oxidation reactions. We also assessed the impact of

450 shipping emissions on HONO formation in coastal regions by carrying out sensitivity experiments. The model results indicated that shipping emissions contributed to regional increases in $NO_x$ and $NO_3^-$ by 17% and 33%, consequently elevating the average HONO concentration by 18%. The contribution of shipping emissions to HONO decreased from coastal to inland areas. The increased HONO concentrations could be explained by the enhanced production rates. Shipping emissions contributed more evidently to HONO production rates during the daytime, with heterogeneous

455 uptake of $NO_2$ on the ground surface, $NO_x$ photo-oxidation, NO+OH, and nitrate photolysis accounting for 39%, 34%, 13%, and 12% of the total enhancement, respectively.

Including good HONO source representation significantly led to the rise in concentrations of OH radicals and $O_3$. The SOM process analysis revealed that photolysis of HONO accounted for 34% of primary OH formation. Meanwhile, this chemical reactive pathway played a more important role in OH formation during the morning rush hours. A sensitivity

460 experiment showed that the diurnal peak of OH levels increased by 61% due to improved representation of HONO sources in the WRF-Chem model. Adjoint SOM analysis of two modeling cases involving BASE and REV further elucidated that only 32% of the increased OH production rate was explained by the direct influence of HONO photolysis; the remainder was contributed by other reactions, including $HO_2$+NO, $O^1D$+$H_2O$, and $H_2O_2$+hv, due to ambient increased AOC conditions. Consequently, the average $O_3$ concentration increased by 44% in the coastal regions of Fujian,

465 a much more significant elevation than in previous studies.

Overall, the present study highlights the critical importance of characterizing HONO's formation mechanisms and environmental impacts in coastal regions. Moreover, our study discussed the potential uncertainties regarding HONO simulations in coastal regions, including those arising from the parameterizations of heterogeneous uptake and nitrate photolysis, the estimates of direct emissions from shipping activities, and the unknown impact of the ocean surface.



Therefore, the representative model parameterizations concerning the complicated formation processes of HONO should be continuously developed and coupled into numerical models in the future.

**Data availability.** The source codes of WRF-Chem model are publicly available on the official website of GitHub (https://github.com/wrf-model/WRF). Meteorological input data for modeling are archived at https://rda.ucar.edu/, with

the product codes of d083002, d461000 and d351000 for FNL reanalysis dataset, surface and upper weather observations. Anthropogenic emission inventories of MEIC and SEIM developed by Tsinghua University can be downloaded from the website of http://meicmodel.org.cn/. Other observational and modeling data used in this study are archived on the Figshare platform at https://doi.org/10.6084/m9.figshare.29827070.v1.

**Author contributions.** XH conceptualized and supervised the study. HRZ performed the model simulations. CYY, SHW

and ELN provided the measurement data in Fujian. HRZ and XH analyzed the data and interpreted the results with the help from TYL. HRZ wrote the original manuscript. XH and HRZ revised and edited this paper with contributions from all other co-authors.

**Acknowledgements.** This study was supported by the Ministry of Science and Technology of the People's Republic of China (2022YFC3701100), the National Natural Science Foundation of China (42293322 and 424B2040), the

Natural Science Foundation of Fujian Province (2022J01518 and 2024J01168) and the Comprehensive Observation and Research Project on Atmospheric Pollution in the Meizhou Bay Region.

**Competing interests.** The contact author has declared that none of the authors has any competing interests.

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
