# Peer review of "HONO Formation Mechanisms and Impacts on Ambient Oxidants in Coastal Regions of Fujian, China"

_EGUsphere, 2025_

## Author Comment (AC1)

**Response to Reviewer #1**

Dear Editor and Reviewer:

We greatly appreciate your consideration and the reviewer's insightful and constructive comments on the manuscript "HONO Formation Mechanisms and Impacts on Ambient Oxidants in Coastal Regions of Fujian, China" (egusphere-2025-2630). We have carefully revised the manuscript to address all the comments described below. Reviewer comments are shown in black. Our responses are shown in blue. The revised texts are shown in red.

Zhang et al. conducted a comprehensive case study in a coastal region to investigate HONO formation mechanism. The diurnal cycle of HONO is well simulated and successfully explained by the model. Results are convincing and sound. The manuscript reads very clear. I think it in general deserves an ACP publication, while I still have a few comments and some minor text-revising suggestions as below:

**Response:** We thank you for the comments. Based on your helpful and insightful comments, we have revised our manuscript, and the point-by-point responses to the specific comments were given subsequently. We sincerely hope these revisions could address your concerns.

**Comments:**

1. It is interesting to observe the peak HONO at noon. However, if I understand correctly, it is from a monthly average in no-raining days. I am wondering if this diurnal pattern is consistent across individual days, or if multiple distinct patterns exist but are masked by the averaging. From a rough inspection of Fig. 2, the diurnal patterns seem to vary from day to day. I think authors should at least demonstrate that the noon-peak HONO occurred on most days. Even better, they are encouraged to do more analysis to examine 1) if model can capture different diurnal patterns and 2) if the underlying mechanisms can also be explained.

**Response:** Thanks for your conducive comments and rigorous attitude to scientific research. We agree with you that averaging can sometimes mask day-to-day variability. In Figure 2, we illustrate the diurnal pattern of HONO concentrations during the entire study period. As you suggested, we have further examined the agreement between observations and simulations on non-rainy and rainy days. As shown in Figure R1, it is revealed that the correlation coefficient (R) between HONO measurements and simulations is 0.75, which is significantly higher than that on rainy days (0.20). The diurnal peak around 14:00 of local time was

reasonably captured by the model on most non-rainy days. However, we noticed that the diurnal pattern of HONO concentrations influenced by precipitation shows less variability. Meanwhile, our model overestimated the observed maximum HONO concentration and failed to reproduce the peak hour. This mismatch between observations and simulations on rainy days may be indicative of an underestimated wet removal effect. On the other hand, the simulated daytime NO2 is in a good agreement with observations under non-raining conditions (Figure R1). But the modelled daytime NO2 concentration is higher than the observed values on the rainy days, plausibly overpredicting chemical production rate of HONO. To make it clearer, we have rephrased the associated texts in the manuscript as follows.

**Figure R1.** The diurnal variations of the observed and simulated HONO and NO2 concentrations on rainy days and non-rainy days.

**Revisions in Section 3.2:**

Figure 3b illustrates that the REV case successfully captured the higher HONO concentrations observed around noon. The Pearson's correlation coefficient (R) between the measurements and simulations increased from 0.657 (BASE) to 0.763 (REV). The REV simulation accurately captured the timing of the observed diurnal peak around 14:00, which the BASE case simulates several hours too early. We further show the comparisons between measurements and simulations in terms of diurnal variations of HONO and NO2 on rainy and non-rainy days. As shown in Figure S3, the daytime HONO concentrations simulated by REV is in good agreement with observations on non-rainy day. However, the model fails to reproduce the observed low concentration levels and the temporal variations of HONO on rainy days, possibly attributed to an underestimated wet removal caused

by precipitation. Meanwhile, daytime NO2 concentrations on rainy days also shows an obvious overestimation, further leading to the higher chemical production of HONO.

2. It might be insufficient to only discuss the HONO chemical production when investigating the impacts of ship emissions on HONO diurnal cycles. I am wondering whether the diurnal cycle of land-sea wind could also play a role in transporting NOx/HONO?

Response: Thank you for this valuable comment. We agree with your point regarding the important role of sea-land breeze circulation in pollutant transport within coastal regions (Huang et al., 2025; Liu et al., 2025; Shen et al., 2021). The sea-land breeze (SLB) is defined as a unique wind pattern in coastal areas where the wind direction reverses between daytime and nighttime due to the differential heating of land and sea. We have identified SLB events during the study period following the criteria used by Liu et al. (2025). Specifically, an effective SLB cycle includes the land breeze phase (01:00-08:00), the sea breeze phase (13:00-20:00), and transition periods in between. In the coastal regions of Fujian, sea breezes are winds from 50° to 220° and land breezes from 240° to 40°. In addition, the sea breeze or land breeze phase should last at least 4 hours, with the opposing breeze lasting no more than 2 hours during these phases. Meanwhile, the wind speed throughout the SLB cycle should be less than 10 m s-1. As illustrated in Figure R2a, our analysis based on the wind field data in Figure 2 reveals that a typical sea-land breeze event occurred on only one day during the entire one-month study period (i.e. 30th of May). A further analysis of atmospheric circulation indicates the coastal region of Fujian was dominated by a persistent northeasterly wind (Figure R2b) for most time, suppressing the formation of local, thermally-driven sea-land circulations, thus limiting their overall role in transport. Overall, the threedimensional numerical model (WRF-Chem) is capable of simulating atmospheric transport processes. Therefore, the net effect of both the dominant synoptic winds and local circulations have been accounted for in our simulation results. To include the discussion about the potential effect of sea-land breeze, we have revised the relevant texts in the manuscript as follows. We would like to express our gratitude to the insightful suggestion from the reviewer again.

**Figure R2.** Wind fields during the study period. Panel (a) shows the temporal variations of wind vectors at the site DZSK. The shaded areas represent the nighttime (19:00 to 6:00). Panel (b) illustrates the spatial pattern of the sea level pressure (hPa) and the surface wind field at 12:00 on each day in May 2024.

**Revisions in Section 3.3.2:**

The impact of the shipping emissions is based on the atmospheric transport of air pollutants from the upstream region. Specifically, the regional transport is driven by both background circulation and local circulations such as sea-land breeze (SLB). Following the criteria of SLB given by Liu et al. (2025), we identified SLB events over the study region based on the local wind field data exhibited in Figure 2. The further analysis of wind fields demonstrates that the study area is less affected by the SLB. The impact of shipping emissions on HONO formation was mainly attributed to the transport effect of regional persistent northeasterly wind.

**Minor suggestions:**

1. Line 20: please specify the model name in the abstract

**Response:** Thank you for pointing this out. We have specified the full name of the numerical model in the revised manuscript.

**Revisions in Abstract:**

Using an updated Weather Research and Forecasting coupled with Chemistry (WRF-Chem) model, we captured the magnitude and temporal variation of HONO concentrations observed in coastal areas, and improved the model performance on diurnal patterns of the NO2 and O3.

2. Lines 39-40 and Lines 48-49: The new reactions of the photo- and dark-oxidation of NOx should also be listed. In the reactions of HONO removal, the HONO oxidized by OH should also be included, as it is used in the following analyse (Fig. 5a).

**Response:** Thanks for your reminder. We have added the missing reactions and rephased the related texts in the manuscript.

**Revisions in Section 1:**

In addition, a recent laboratory study also reported that the photo- and dark-oxidation of nitrogen oxides  $(NO_x)$  could significantly contribute to HONO formation (R4-R5) (Song et al., 2023).

In the presence of sunlight, HONO can rapidly undergo photodissociation (R6) to yield NO and OH radicals (Seinfeld and Pandis, 2016). Meanwhile, HONO can be slightly consumed by OH oxidation (R7).

| $\underline{HNO_3} + NO \rightarrow HONO + NO_2$ | (R4) |
|--------------------------------------------------|-------------|
| $NO_3 + NO \rightarrow 1.98 NO_2 + 0.02 HONO$    | (R5) |
| $\underline{HONO} \rightarrow OH + NO$           | (R6) |
| $HONO + OH \rightarrow NO_2 + H_2O$              | (R7) |

3. Lines 56-58: what are the associated conclusions of this previous study?

**Response:** Thank you for pointing this out. A new description has been rephrased in the relevant paragraph.

**Revisions in Section 1:**

Our previous study focused on the Yangtze River Delta (YRD) region indicates that HONO contributed to a 22% and 28% increase in concentrations of fine particulate matter and O3 during a short-term compound air pollution case in March of 2019 (Zhang et al., 2024b).

4. Lines 102-103: It is not clear what is the 'seven-day loop cycle'? What 'systemic

biases' authors want to avoid?

**Response:** Thanks for this comment. The seven-day loop cycle we used refers to manually restarting the WRF-Chem model every seven days. The objective of this process is to reduce the long time accumulation of biases of numerical computation. To clarify this point, we have revised the associated texts in the manuscript as follows.

**Revisions in Section 2.2:**

The entire WRF-Chem simulation was manually restarted every seven days to reduce the influence of the accumulation of biases from the numerical computation.

5. Table 2. Please clarify that the 120 is from the ratio between measured NO3- photolysis rates and the default HNO3 photolysis rate.

**Response:** Thanks for your kind reminder. We have corrected this point as you suggested.

**Revisions in Table 2:**

Table 2. Chemical parameterizations of HONO sources in the revised WRF-Chem model.

| Pathway        | Parametrization                                  | Descriptions                             |
|-----------------------|---------------------------------------------------------|-------------------------------------------------|
| Photolysis of nitrate | $\underline{J_{NO3}} = 120 \times \underline{J_{HNO3}}$ | $\underline{J_{NO3}}$ - and $J_{HNO3}$ are the  |
| aerosols       |                                                         | photolysis frequencies of nitrate               |
|                       |                                                         | aerosols and gaseous HNO3 (Fu                   |
|                       |                                                         | et al., 2019). The ratio of 120 is              |
|                       |                                                         | estimated between the measured                  |
|                       |                                                         | NO3 photolysis rate and the                     |
|                       |                                                         | default HNO3 photolysis rate. |

6. Line 215. Figure 3b? Meanwhile, it looks like the BASE experiment also shows noon-peak HONO. Could you explain why it is? Could it be possible to imply that the diurnal emissions/transport also contribute to the noon-peak HONO?

**Response:** Thank you for your careful comment. The typo of the figure number has been corrected in our revised manuscript. The BASE scenario refers to the default WRF-Chem model which only takes the homogeneous reaction between NO and OH into account. The direct emission of HONO is turned off in BASE so that direct emissions and transport from upstream regions could not contribute to local HONO formation. In BASE scenario, we believe that the noon-peak of HONO concentrations is attributed to the higher production rate of the reaction NO + OH (Figure R3). Both concentrations and production rates of HONO reach their peaks at 11:00 of local time. The NO + OH reaction rate is highest at noon because

OH radical concentrations, driven by photochemistry, peak at this time. This explains the timing of the noon-peak in the BASE case. However, only considering the reaction between NO and OH fails to capture well the chemical production fate of HONO during the daytime that the daily maximum concentration of HONO in BASE is significantly lower than the observed value.

**Figure R3.** The simulated concentrations and production rates of HONO in BASE scenario which only considers homogenous reaction between NO and OH.

7. Lines 314-316. Which one is daytime and which one should be nighttime?

**Response:** Thank you for your careful comment. This typo has been corrected as follows.

**Revisions in Section 3.3.2:**

However, due to the self-photolysis of HONO, the overall increase in coastal HONO concentrations caused by shipping emissions during the daytime (0.19 ppbv) was close to nighttime levels (0.20 ppbv).

8. Lines 422-423. Please specify the ratio of HONO to NOx of mobile sources are supposed to be higher or lower?

**Response:** Thanks for your insightful suggestion. According to the recent report by Ke et al. (2025), the ratio of HONO to NOx of mobile sources are expected to be higher. For aircraft, this ratio ranges from 0.82% to 6%. For offroad mobile machinery, this ratio can reach 2.3%. To specify this point, we have rephrased the related texts in the manuscript as you suggested.

**Revisions in Section 3.5:**

However, Ke et al. (2025) have pointed out that the same ratio of HONO to  $NO_x$  may not be applicable to mobile sources, including mobile machinery, ships, and aircraft. The ratios of HONO to  $NO_x$  were examined to be  $0.8\sim6.0\%$  and 2.3% for

**aircraft and offroad mobile machinery, respectively.**

9. The uncertainty of soil HONO emissions should also be discussed, especially there are large cropland areas in the study domain.

**Response:** Thank you for pointing this out. We indeed agree with you that the uncertainty of direct emissions from soils should be discussed. We have incorporated the related discussions in the manuscript.

**Revisions in Section 3.5:**

In addition to mobile emissions for HONO from combustion processes, the soil volatilization could be another important contributor to ambient HONO (Tan et al., 2023; Wu et al., 2022), especially when there are large areas of cropland and agricultural activities. The neglect of soil HONO emissions may introduce some uncertainties in the HONO simulation in this study.

**References**

- Fu, X., Wang, T., Zhang, L., Li, Q., Wang, Z., Xia, M., Yun, H., Wang, W., Yu, C., Yue, D., Zhou, Y., Zheng, J., and Han, R.: The significant contribution of HONO to secondary pollutants during a severe winter pollution event in southern China, Atmospheric Chem. Phys., 19, 1–14, https://doi.org/10.5194/acp-19-1-2019, 2019.
- Huang, Y., Li, S., Zhu, Y., Liu, Y., Hong, Y., Chen, X., Deng, W., Xi, X., Lu, X., and Fan, Q.: Increasing Sea-Land Breeze Frequencies Over Coastal Areas of China in the Past Five Decades, Geophys. Res. Lett., 52, e2024GL112480, https://doi.org/10.1029/2024GL112480, 2025.
- Ke, J., Yang, X., Lu, K., Fu, M., Wang, Y., Yin, H., and Ding, Y.: Overlooked Underestimation of Mobile Sources Posing a Pronounced Imbalance in the HONO Budget, Environ. Sci. Technol., 59, 5875–5877, https://doi.org/10.1021/acs.est.5c02684, 2025.
- Liu, C., Wang, H., Li, L., Chen, X., Lu, X., and Fan, S.: Impacts of sea-land breeze on the coastal ozone in the Pearl River Delta, China, J. Environ. Sci., https://doi.org/10.1016/j.jes.2025.08.037, 2025.
- Shen, L., Zhao, C., and Yang, X.: Climate-Driven Characteristics of Sea-Land Breezes Over the Globe, Geophys. Res. Lett., 48, e2020GL092308, https://doi.org/10.1029/2020GL092308, 2021.
- Tan, W., Wang, H., Su, J., Sun, R., He, C., Lu, X., Lin, J., Xue, C., Wang, H., Liu, Y., Liu, L., Zhang, L., Wu, D., Mu, Y., and Fan, S.: Soil Emissions of Reactive Nitrogen Accelerate Summertime Surface Ozone Increases in the North China Plain, Environ. Sci. Technol., 57, 12782–12793, https://doi.org/10.1021/acs.est.3c01823, 2023.
- Wu, D., Zhang, J., Wang, M., An, J., Wang, R., Haider, H., Xu-Ri, Huang, Y., Zhang, Q., Zhou, F., Tian, H., Zhang, X., Deng, L., Pan, Y., Chen, X., Yu, Y., Hu, C., Wang,

R., Song, Y., Gao, Z., Wang, Y., Hou, L., and Liu, M.: Global and Regional Patterns of Soil Nitrous Acid Emissions and Their Acceleration of Rural Photochemical Reactions, J. Geophys. Res. Atmospheres, 127, e2021JD036379, https://doi.org/10.1029/2021JD036379, 2022.

---

## Author Comment (AC2)

**Response to Reviewer #2**

Dear Editor and Reviewer:

We greatly appreciate your consideration and the reviewer's insightful and constructive comments on the manuscript "HONO Formation Mechanisms and Impacts on Ambient Oxidants in Coastal Regions of Fujian, China" (egusphere-2025-2630). We have carefully revised the manuscript to address all the comments described below. Reviewer comments are shown in black. Our responses are shown in blue. The revised texts are shown in red.

This paper conducted a one-month HONO observation at a suburban site in coastal Fujian, combined with an improved WRF-Chem simulation, to systematically investigate the mechanism of high noontime HONO and the impact of shipping emissions on regional oxidizing capacity. The overall idea is complete, with close integration of observation and simulation, and the results are of great significance for a deep understanding of the HONO source mechanism in coastal areas and for quantifying the contribution of shipping emissions to regional atmospheric oxidizing capacity. However, this study still has many shortcomings in model settings, discussion depth and expression, and the authors need to carefully revise the paper to ensure the reliability and rationality of the results. The specific comments are as follows:

**Response:** We thank you for the comments. Based on your helpful and insightful comments, we have revised our manuscript, and the point-by-point responses to the specific comments were given subsequently. We sincerely hope these revisions could address your concerns.

1. Section 2.1 states that the observation site is about 25 km from the Taiwan Strait. However, during the daytime, HONO quickly dissipates due to photolysis, with a typical lifetime of only a few tens of minutes. The researchers also reported low wind speeds during the observation period (average WS = 2.1 ms-1). A simple transport calculation gives: the transport time from the ocean to the observation site is 3.3 hours (t = 25000/2.1/3600), which is one order of magnitude longer than the daytime HONO lifetime. Under this condition, any HONO directly emitted over the Strait is expected to decay significantly before reaching the receptor site. Therefore, the paper may overestimate the contribution of daytime shipping emissions to observed HONO concentrations and atmospheric oxidizing capacity.

Response: We thank the reviewer for this insightful comment. We agree with your point that the simple calculation correctly shows that due to short photochemical lifetime of 10-20 minutes, HONO cannot undergo direct long-range transport from the Taiwan Strait to the local observation site. We would like to clarify that our central argument is not the transport of HONO itself, but rather that of its more stable precursors, mainly nitrogen oxides (NOx). Our model simulates the process wherein NOx emitted from shipping activities is transported to the coastal areas. Upon arrival, this precursor-rich airmass undergoes rapid chemical conversion to HONO (light-enhanced heterogeneous reactions and photo-oxidation reactions), contributing to the observed high production rates during the daytime. Therefore, the contribution from shipping emissions is mainly realized through the transport of precursors followed by local formation, a mechanism consistent with the short lifetime of HONO. To clarify this point, we have revised the manuscript as follows.

**Revisions in Section 3.3.2:**

It is worth noting that the contribution from shipping emissions to coastal HONO formation is mainly driven by the transport of precursors including NOx and NO3. That is, shipping emissions affect daytime HONO formation via precursor transport followed by local chemical production despite HONO's short atmospheric lifetime.

2. L89-L90: NO2 was measured using 17i, also using chemiluminescence, which will overestimate NO2 concentration and should be corrected. In addition, the concentration units of the same species in the manuscript should be unified. Was the concentration of NO, an important precursor of HONO, measured? Why is it not shown?

Response: Thank you for pointing this out. We acknowledge the point that standard chemiluminescence analyzers with molybdenum converters (such as the Thermo 17i) can have a positive artifact from other reactive nitrogen species, potentially leading to an overestimation of the true NO2 concentration (Dunlea et al., 2007). No specific correction was applied to the data in our study, and we have added a cautionary note regarding this uncertainty to the Methods section. We have performed a careful check of the entire manuscript and unified the concentration units for all species to ensure consistency. Regarding NO, this species was measured during the field campaign; however, the amount of valid data was insufficient for a robust analysis, and therefore it is not presented in this study.

**Revisions in Section 2.1:**

For mode 17i, we acknowledge that the chemiluminescence instrument with a molybdenum converter used for NO2 measurements may be subject to positive artifacts from other reactive nitrogen species, which represents a potential source of uncertainty in the NO2 data used for model evaluation (Dunlea et al., 2007).

3. L134-136: In most studies the NO2 uptake coefficient on the ground is smaller than that on the aerosol surface. The authors should provide sufficient reasons for this choice. In addition, the light-enhanced NO2 uptake coefficient is generally on the order of 10-5, and in some studies 1×10-3 has only been used as the upper limit of NO2 heterogeneous reactions. This value will seriously overestimate the contribution of NO2 heterogeneous reactions, and it is recommended that the authors reconsider the value. In addition, selecting 1.45% as the emission factor is also significantly higher than the commonly used 0.8%. The authors should calculate the corresponding emission factor based on field observations to increase the rationality of the value.

Response: We thank the reviewer for the careful comment on these key heterogeneous uptake coefficients. Our choice of parameterization is in accordance with several previous modeling studies conducted in China (Zhang et al., 2021; Liu et al., 2019a; Zhang et al., 2024). To be specific, under dark conditions, we set the NO2 uptake coefficients to  $8\times10^{-6}$  for the ground surface and  $4\times10^{-6}$  for aerosol surfaces. For daytime, these values were dynamically scaled with solar radiation, reaching their maximums only under the strongest sunlight conditions at  $6\times10^{-5}$  for the ground surface and  $1\times10^{-3}$  for aerosol surfaces. We acknowledge that the value of 1×10-3 represents an upper limit for the light-enhanced NO2 uptake on aerosol surfaces as reported in the literature. However, the critical factor in this coastal study area is the low ambient aerosol concentrations. This limited availability of aerosol surface area means that even with a high uptake coefficient, the overall contribution of this pathway to HONO formation is minimal. While PM2.5 mass concentration was not measured during this campaign, our model simulates a regional average PM2.5 concentration of approximately 11.9 μg m-3, proposing a relative clean condition. Simultaneously, the WRF-Chem model reveals that the contribution from heterogeneous reactions on aerosol surfaces accounted for a negligible 2% of the total daytime HONO production (Figure 5). Therefore, we are confident that this parameter does not overestimate the HONO budget in the present study. We have rephrased the relevant texts in the revised manuscript to

clarify the rationale behind these choices.

Regarding the HONO/NOx emission ratio, we used a value of 1.45% in this study, which is higher than the more widely adopted value of 0.8% (Kurtenbach et al., 2001). Our choice of 1.45% is based on the estimates of Hu et al. (2022), which derived this ratio from a long-term measurement campaign in Xiamen, a coastal city also located in the study region. We agree with the reviewer that deriving a constrained emission ratio directly from our own field observations would be the most robust approach. However, characterizing the direct emission ratio requires a long-term dataset (typically several months to a year) to collect sufficient fresh emission plumes (Liu et al., 2019b). As our measurement campaign was limited to a one-month period, we were unable to derive a statistically robust HONO/NOx ratio from our dataset. Thus, we consider the ratio proposed by Hu et al. from a nearby location to be an appropriate alternative.

**Revisions in Section 2.3:**

The NO2 heterogeneous uptake on ground and aerosol surfaces was parameterized as a light-dependent process, with uptake coefficients ( $\gamma$ ) chosen in accordance with previous studies in China (Zhang et al., 2021; Liu et al., 2019a; Zhang et al., 2024). The base nighttime uptake coefficients of NO2 were set to  $8\times10^{-6}$  for the ground surface and  $4\times10^{-6}$  for aerosol surfaces. During the daytime, these values were dynamically increased with solar radiation using a linear equation, reaching their maximums of  $6\times10^{-5}$  and  $1\times10^{-3}$ , respectively (Liu et al., 2019).

4. The IOA index increased from 0.62 (BASE) to 0.69 (REV), which is not very high. At the same time, in Fig. 3a and 3b, the fit of the REV simulation results with the observations is poor, and the simulated values are significantly higher than the observed values on many days. These simulation results are difficult to convince readers. Did the authors consider the effect of rainy days when calculating the model evaluation index? The authors did not clearly state this. In the diurnal variation diagram of Fig. 3b, why are the three curves shifted?

**Response:** We would like to thank the reviewer for the detailed feedback on our model evaluation. We indeed agree with you that the improvement of IOA index from 0.62 to 0.69 is modest. As shown in Figure 3b, the BASE simulation, which only includes the homogeneous reaction of NO+OH, is also able to produce a midday HONO peak. However, a key deficiency in the BASE case is that this peak occurs much earlier than observed. The revised HONO model could more

accurately captures the timing of the observed peak concentration around 14:00. More importantly, the BASE case failed to reproduce the observed magnitude of HONO concentrations. This improvement in REV is better reflected by NMB and RMSE. Regarding the systematic overestimation from 19th to 25th of May, this period corresponds to continuous rainfall, during which model is likely affected by uncertainties from wet scavenging. Our current evaluation in Table 3 is based on the entire month to provide an assessment. To address your point, we have conducted an additional evaluation using a dataset filtered for non-rainy conditions only to better demonstrate the model's performance under typical dry conditions. Concerning the "shift" of the curves in Figure 3b, this presentation style was chosen to give a direct comparison between observations and simulations. To improve clarity for the reader, we have revised the figure's presentation as follows.

**Revisions in Section 3.2:**

As summarized in Table 3, while the improvement in the Index of Agreement (IOA, varies from 0 to 1) is modest (from 0.62 to 0.69), the revised model shows a fundamental improvement in capturing the magnitude of HONO concentrations. This is demonstrated by the dramatic enhancements in the Normalized Mean Bias (NMB, varies from  $-\infty$  to  $+\infty$ ), which improved from -86% to +8%, and the Root Mean Square Error (RMSE, varies from  $-\infty$  to  $+\infty$ ), which decreased by 21%. While the revised model reasonably reproduced the observed temporal variations in HONO concentrations during the study period, an underestimation existed on 16-18 May, suggesting a potential omission of HONO sources. The systematic overestimation during 21-25 May corresponds to a period of continuous rainfall. To provide an evaluation focused on the normal conditions, we also calculated the statistics for non-rainy periods only, where the model performance improved further (IOA = 0.70, NMB = -5%, RMSE = 0.21 ppbv).

Figure 3b illustrates that the REV case successfully captured the higher HONO concentrations observed around noon. The Pearson's correlation coefficient (R) between the measurements and simulations increased from 0.657 (BASE) to 0.763 (REV). The REV simulation accurately captured the timing of the observed diurnal peak around 14:00, which the BASE case simulates several hours too early.

**Revisions in Figure 3b:**

5. Why do the authors not consider the removal pathway of HONO deposition, especially since nighttime HONO removal is mainly the deposition process.

Response: Thank you for this insightful comment. This is an excellent point. We fully agree that dry deposition is an important sink for HONO, particularly during nighttime. In our source-oriented method (SOM) analysis, the focus was specifically on quantifying the contributions from various chemical production and loss pathways, which is why deposition was not explicitly tracked as a sink in the budget analyses (Grell et al., 2005). However, the dry deposition process for HONO and other species is indeed calculated online within the standard WRF-Chem framework and contributes to the overall simulated concentrations. We have rephrased the relevant texts in the revised manuscript to acknowledge the importance of deposition as a nighttime sink for HONO.

**Revisions in Section 2.3:**

We also quantified two HONO chemical sink pathways: photodissociation of HONO (HONO+hv) as well as OH-oxidation removal (HONO+OH). Additionally, it should be noted that dry deposition, an important sink for HONO especially at night, is calculated within the standard WRF-Chem deposition module but was not explicitly tracked in this chemical budget analysis since our focus here was on chemical pathways.

6. The authors should provide PM2.5 concentrations to support the conclusion that NO2 heterogeneous reactions on the aerosol surface contribute little.

**Response:** Thank you for this constructive suggestion. We do agree with you that PM2.5 data would strengthen our conclusion. While PM2.5 mass concentration was

not measured during this campaign, our model simulates a regional average  $PM_{2.5}$  concentration of approximately  $11.9 \,\mu g \, m^{-3}$ , which is significantly lower than those typical concentrations in inland regions of China. We have added this simulated value and a brief discussion to the revised manuscript to support our point that the contribution from the heterogeneous uptake of  $NO_2$  on aerosol surfaces is limited in clean coastal environment.

**Revisions in Section 3.3.1:**

Similarly, the contribution from the heterogeneous NO2 uptake on aerosol surfaces (1–2%) was lower than that reported for inland areas (3–20%), because of lower particle concentrations in coastal regions. The WRF-Chem model shows that the average PM2.5 concentration over the coastal areas of Fujian was 11.9 µg m-3 during the study period, which is categorized into the clean state and is much lower than the levels in typical inland regions.

7. In the updated HONO sources, the parameter values should be explicitly provided or the calculation process shown. For example, how were the S/V of ground and aerosol surfaces calculated?

**Response:** Thanks for your careful reminder. We have added the illustration of calculating the key parameter surface area density (S/V) for the ground surface and aerosol surfaces in the method section as you suggested.

**Revisions in Section 2.3:**

 $S_a/V$  and  $S_g/V$  are aerosol and ground surface area densities ( $m^2 \, m^{-3}$ ), respectively.  $S_a/V$  could be calculated through the MOSAIC aerosol module, which categorized different types of aerosols into four size bins ranging from 3.9 nm to 10 µm, i.e. 0.039-0.156 µm, 0.156-0.625 µm, 0.625-0.2500 µm and 0.2500-0.000 µm (Zaveri et al., 0.208). 0.2080. 0.2081. 0.2082 was derived based on the underlying surface category. In vegetation grid cells, 0.2082 was estimated as the ratio of the two-fold of leaf area index (LAI, 0.2082 m² to the model height of the first layer (Zhang et al., 0.2083. For urban areas, the ground surface area density 0.2084 was empirically set from 0.14 to 0.2085 depending on the fraction of urban area using a linear formula (Zhang et al., 0.2084). It is noted that the model only accounts for heterogeneous uptake of 0.2085 on ground surface in the first layer, while the reaction on aerosol surfaces occurs in all model layers.

8. In Section 3.3.2 the authors explain "...meaning that shipping emissions contributed less to coastal NOx during the daytime." However, the daytime HONO production rate is relatively high. In theory, as an important precursor of HONO, if the impact of NOx from shipping emissions is low, even if there are lightenhanced reactions, the HONO production rate should be limited. Therefore, the high daytime HONO production rate cannot be explained by "light-dependent reaction pathways." At the same time, the explanation in Section 3.3.3 is also not valid.

**Response:** Thanks for your conducive comments and we acknowledge the need for a clearer explanation. While the relative contribution of shipping emissions to the total NOx concentration is lower during the daytime, the absolute concentration of NOx from both shipping and continental sources remains sufficient to fuel HONO production. The dramatic increase in the HONO production rate is driven by the enhanced efficiency of light-dependent pathways. Therefore, the high production rate is a consequence of sufficient precursor availability combined with high photochemical conversion efficiency. We have rephrased the relevant texts in the revised manuscript to for clarity.

**Revisions in Section 3.3.3:**

The captured high HONO concentrations over the study region between 11:00 and 14:00 were attributed to the increase in chemical production rates (see Figures 5a and 7a). There are two main factors. One is a sufficient supply of NOx precursors from both continental and shipping emission sources that, even while being at a diurnal minimum around noon, remains ample to fuel the subsequent reactions. The other is an enhanced reaction rate of light-dependent pathways under intense solar radiation.

9. Sensitivity analysis was not sufficiently carried out. The authors should scale the various parameters used by a certain proportion and then analyze how this parameter change affects the contribution of HONO sources or the impact on OH/O3 concentrations. The uncertainty analysis in Section 3.5 is not an explanation of the reasons for the parameter values, but should involve sensitivity experiments for the parameter values and discussion of their impact on HONO production rate, OH and O3.

**Response:** We thank the reviewer for this constructive suggestion. Among those parameters used in this study, the HONO/NOx ratio was based on long-term

observations in Fujian and was representative (Hu et al., 2022). Similarly, our parameterizations for heterogeneous NO2 uptake on the ground surface (varying between 10-6 to 10-5) and nitrate photolysis were set to robust and median-level values widely used in previous studies (Fu et al., 2019; Wang et al., 2025; Zhang et al., 2021). This leaves the uptake coefficient of NO2 on aerosol surfaces ( $\gamma_a$ ) as the parameter with the largest uncertainty in our scheme, for which we adopted an upper-limit value to represent the maximum light-enhanced process. Following your suggestion, we have conducted two additional sensitivity experiments to quantitatively assess the impact of this highly uncertain parameter. Specifically, we reduced the maximum daytime  $\gamma_a$  by one and two orders of magnitude, respectively. To balance computational cost and storage, these new simulations were performed for the first seven days of our study period. Model results show that while lowering the  $\gamma_a$  value does lead to a corresponding decrease in the HONO production rate from the heterogeneous uptake of NO2 by aerosols, the impact on the overall HONO budget is negligible. The average daytime HONO concentration decreased by less than 2 pptv, a relative change of less than 1%. This finding provides quantitative support for our argument that due to the low aerosol abundance in this coastal region, the heterogeneous aerosol pathway contributes minimally to HONO formation, regardless of the precise γa value. Consequently, the responses in O3 and OH concentrations were also minimal. This confirms that our use of  $1 \times 10^{-3}$  as an upper-limit for  $\gamma_a$  is a reasonable choice and does not compromise the main conclusions of our study. We have incorporated this new analysis into the revised manuscript. We thank you again for this valuable suggestion.

**Table R1.** Influences of different  $\gamma_a$  on daytime production rates, concentrations of HONO, and concentrations of ambient oxidants.

| Case  | Maximum daytime γ a | Production rate
from Hete_NO 2
on aerosols | HONO (ppbv) | O 3 (ppbv) | OH (×10 6 molecules cm -3 ) |
|-------|--------------------------------|-------------------------------------------------------------|-------------|-----------------------|---------------------------------------------------|
|       | any cirrio qu                  | (ppbv h -1 )                                     | (PP° ·)     | (PP°)                 | , , , , , , , , , , , , , , , , , , ,             |
| REV   | 1×10 -3             | 0.0156                                                      | 0.223       | 40.3                  | 6.1                                               |
| Sens1 | 1×10 -4             | 0.0016                                                      | 0.221       | 40.4                  | 6.1                                               |
| Sens2 | 1×10 -5             | 0.0002                                                      | 0.221       | 40.4                  | 6.1                                               |

**Revisions in Section 3.5:**

Several uncertainties exist in the HONO simulations presented in this study, firstly related to the parameterization of key chemical pathways. These are mainly concentrated in the heterogeneous uptake coefficients and the nitrate aerosol photolysis rate. For the nitrate photolysis frequency, Zhang et al. (2022)

summarized that this value is approximately 1-3 orders of magnitude higher than the photolysis frequency of HNO3. Our study adopted a median value of this range (120JHNO3), which was inferred based on aircraft measurements in the North Atlantic marine boundary layer and has been widely used in previous studies (Fu et al., 2019; Ye et al., 2016; Zhang et al., 2021). For the heterogeneous uptake of NO2 on solid surfaces, the dimensionless uptake coefficient typically ranges from 10-6 to 10-3. For the ground surface, we also applied a representative median value, with this coefficient varying from a nighttime baseline in the 10-6 range up to a maximum of  $6 \times 10^{-5}$  under peak sunlight (Wang et al., 2025). For the aerosol surfaces, we set the maximum daytime value to  $1 \times 10^{-3}$ , an upper limit reported in the literature, which carries a potential uncertainty. To quantitatively assess the impact of this choice, we conducted an additional sensitivity analysis where the uptake coefficient  $y_a$  was reduced to  $1 \times 10^{-4}$  and  $1 \times 10^{-5}$ , respectively. To minimize the computational burden, these simulations were performed for the first seven days of our study period. As shown in Table S4, model results show that while lowering the  $y_a$  value does lead to a corresponding decrease in the HONO production rate from the heterogeneous uptake of NO2 by aerosols, the impact on the overall HONO budget is negligible. The average daytime HONO concentration decreased by less than 2 ppty, a relative change of less than 1%. This finding provides quantitative support for our argument that due to the low aerosol abundance in this coastal region, the heterogeneous aerosol pathway contributes minimally to HONO formation, regardless of the precise  $\gamma_a$  value. Consequently, the responses in O3 and OH concentrations were also minimal. This confirms that our use of  $1 \times 10^{-3}$  as an upper-limit for  $\gamma_a$  is a reasonable choice and does not compromise the main conclusions of our study.

10. L373-L374: After adding HONO sources in the model, the daytime maximum OH concentration increased to 12.1×106 molecules cm-3, significantly higher than OH concentrations observed in southern China in May, which further challenges the rationality of the parameter values in the updated HONO parameterization scheme. It also shows that the enhancement effect of HONO on O3 in this study is significantly higher than previously reported ranges, which should also be considered in terms of the rationality of the parameters used.

**Response:** We thank the reviewer for this critical point. We acknowledge that the lack of direct OH radical measurements at our site in Fujian prevents a direct

validation of the simulated concentrations. Following your suggestion, we have reviewed previous observational studies of OH radicals in China to provide context for our modeling results. A comparison conducted by Ma et al. (2022) summarized five systematic OH radical measurement campaigns across major polluted regions in China, including the North China Plain, the Yangtze River Delta, and the Pearl River Delta. As shown in Figure R1, their results show that the observed noon-time OH concentrations range from  $4 \times 10^6$  molecules cm-3 to  $13 \times 10^6$  molecules cm-3. Our simulated daily maximum OH concentration ( $12.1 \times 10^6$  molecules cm-3) falls near the upper end of this observed range. While this comparison suggests our simulated value is not outside the range of concentrations measured in other photochemically active environments in China, we agree that further validation against local, in-situ measurements is essential to assess the reasonableness of the updated HONO parameterization scheme. We have revised the manuscript to include this important discussion.

**Figure R1.** Summary of OH radical concentrations (noontime, 11:00–13:00) measured in five summer field campaigns in China. Yufa (YF) and Wangdu (WD) campaigns in the North China Plain, Heshan (HS) and Backgarden (BG) campaigns in the Pearl River Delta, and Taizhou (TZ) campaign in Yangtze River Delta. The box—whisker plot shows the 90th, 75th, 50th, 25th, and 10th percentile values of noon OH radical concentrations in each campaign. The diamond shows the mean values of noon OH radical concentrations. This figure was directly obtained from Ma et al. (2022).

Regarding the O3 enhancement, while the relative increase (44%) is high, we emphasize that the absolute increase (~9.9 ppbv) is consistent with many previous studies (Table 4). The high relative increase is attributed to the fact that the BASE

case severely underestimated O3 concentrations, leading to a very low baseline. The REV case corrects this bias and makes O3 levels much closer to observations, highlighting the critical role of HONO chemistry in this coastal environment. To make this point clearer, we have added a discussion in the revised manuscript.

**Revisions in Section 3.4.1:**

Concurrently, the daily maximum OH concentration rose from 7.5×106 molecules cm-3 (BASE) to 12.1×106 molecules cm-3 (REV). Measurements of OH radicals were not available in this study for a direct model validation. However, a comprehensive study presenting OH measurements from five field campaigns in China reported that observed noon-time peak OH concentrations range from 4×106 molecules cm-3 to 13×106 molecules cm-3 across the NCP, YRD, and PRD regions (Ma et al., 2022). The daily maximum OH concentration simulated in our study falls near the upper end of this observed range. While this suggests our simulated value is within the scope of previously measured concentrations in other photochemically active regions of China, we acknowledge that future validation with local measurements is crucial to fully confirm the reasonableness of the updated HONO chemistry.

**Revisions in Section 3.4.2:**

While the relative enhancement of 44% appears high, it is largely a consequence of correcting the significant underestimation of  $O_3$  in the BASE simulation. The absolute increase is in line with the values reported by many previous modeling studies, emphasizing the importance of including complete HONO sources as possible in 3D models to accurately simulate coastal  $O_3$ .

11. The authors quantified the increments of HONO, NOx, and NO3- from shipping emissions, but there is a lack of spatial comparison analysis with actual shipping routes/port areas. It is recommended to add route or port distribution maps in the SI, and group the analysis by wind direction, to explore the modulation effect of nearshore O3 return/reaction on HONO and NOx.

**Response:** Thanks for this constructive comment. This is an excellent suggestion to improve our analysis. In the revised manuscript, we have added a map of the major shipping routes and ports (please refer to Figure R2), which was obtained from the team at Tsinghua University who developed the shipping emission inventory model (Wang et al., 2021).

**Figure R2.** The spatial distribution of shipping route network and major ports around China. The figures next to the shipping route arcs are the geodesic distances calculated from the ArcGIS tool. This map was directly obtained from Wang et al. (2021).

Regarding the nearshore O3 return, we consider that this process is primarily driven by local sea-land breeze (SLB) circulation. Following the criteria from Liu et al. (2025), we performed an analysis to identify SLB events during our study period. Specifically, an effective SLB cycle requires distinct land (01:00-08:00) and sea (13:00-20:00) breeze phases, with region-specific directional criteria (sea breezes from 50°-220°; land breezes from 240°-40° for coastal Fujian), minimum duration requirements ( $\geq 4$  hours), and exclusion of strong synoptic winds ( $\geq 10 \text{ m s}^{-1}$ ). As illustrated in Figure R3a, our analysis reveals that a classic SLB event occurred on only one day during the entire one-month study period (May 30th). A further analysis of the large-scale atmospheric circulation (Figure R3b) confirms that for most of the period, the coastal region of Fujian was dominated by a persistent northeasterly synoptic flow. This synoptic pattern suppressed the formation of local, thermally-driven circulations, thus limiting their overall role in transport. This analysis directly addresses the reviewer's suggestion to group the analysis by wind direction. It demonstrates that the dominant transport regime during our study was a consistent synoptic flow, not a recurring local SLB circulation. Our WRF-Chem model simulates these atmospheric transport processes, meaning the net effect of both the dominant synoptic winds and any intermittent local circulations is inherently accounted for in our monthly average results. Therefore, our monthly

scale assessment is representative of the prevailing conditions, and a separate, detailed analysis focusing only on the single anomalous SLB day would not be representative of the entire period. To make these points clearer, we have incorporated this analysis into the revised manuscript. We would like to express our gratitude again for this insightful suggestion.

**Figure R3.** Wind fields during the study period. Panel (a) shows the temporal variations of wind vectors at the site DZSK. The shaded areas represent the nighttime (19:00 to 6:00). Panel (b) illustrates the spatial pattern of the sea level pressure (hPa) and the surface wind field at 12:00 on each day in May 2024.

**Revisions in Section 3.3.2:**

The impact of the shipping emissions is based on the atmospheric transport of air pollutants from the upstream region. Specifically, the regional transport is driven by both background circulation and local circulations such as sea-land breeze (SLB). Following the criteria of SLB given by Liu et al. (2025), we identified SLB events over the study region based on the local wind field data exhibited in Figure

- 2. The further analysis of wind fields demonstrates that the study area is less affected by the SLB. The impact of shipping emissions on HONO formation was mainly attributed to the transport effect of regional persistent northeasterly wind.
- 12. Some minor errors: L45 "organic volatile organic compounds (VOCs)" is incorrect; L107 misstates, not Fig. 2b; where is Fig. 3c; L349-L350 and L363-L365 both mention the average daily OH radical production rate, but the values are completely different. The authors should carefully check and distinguish them. **Response:** We sincerely thank the reviewer for the careful reading and pointing out these errors. We have made the following corrections: (1) "organic volatile organic compounds" has been corrected to "volatile organic compounds"; (2) the reference to Fig. 2b has been corrected to Fig. 1b; (3) the reference to Fig. 3c is a typo and has been corrected to Fig. 3b; (4) Regarding the two different OH production rates, we have clarified the text to explicitly state that the value of 2.61 ppbv h-1 represents the total OH production rate, which includes the dominant secondary conversion from HO2+NO, while the value of 1.52 ppbv h-1 represents the average rate from primary sources only during the daytime.

**Revisions in Section 3.4.1:**

Generally, the average daytime production rate of OH from primary sources in the coastal regions of Fujian was estimated to be  $1.52 \text{ ppbv } h^{-1}$ .

**References**

- Dunlea, E. J., Herndon, S. C., Nelson, D. D., Volkamer, R. M., Martini, F. S., Sheehy,
  P. M., Zahniser, M. S., Shorter, J. H., Wormhoudt, J. C., Lamb, B. K., Allwine, E.
  J., Gaffney, J. S., Marley, N. A., Grutter, M., Marquez, C., Blanco, S., Cardenas,
  B., Retama, A., Villegas, C. R. R., Kolb, C. E., Molina, L. T., and Molina, M. J.:
  Evaluation of nitrogen dioxide chemiluminescence monitors in a polluted urban environment, Atmos Chem Phys, 2007.
- Fu, X., Wang, T., Zhang, L., Li, Q., Wang, Z., Xia, M., Yun, H., Wang, W., Yu, C., Yue, D., Zhou, Y., Zheng, J., and Han, R.: The significant contribution of HONO to secondary pollutants during a severe winter pollution event in southern China, Atmospheric Chem. Phys., 19, 1–14, https://doi.org/10.5194/acp-19-1-2019, 2019.
- Grell, G. A., Peckham, S. E., Schmitz, R., McKeen, S. A., Frost, G., Skamarock, W. C., and Eder, B.: Fully coupled "online" chemistry within the WRF model, Atmos. Environ., 39, 6957–6975, https://doi.org/10.1016/j.atmosenv.2005.04.027, 2005.
- Hu, B., Duan, J., Hong, Y., Xu, L., Li, M., Bian, Y., Qin, M., Fang, W., Xie, P., and Chen, J.: Exploration of the atmospheric chemistry of nitrous acid in a coastal city of southeastern China: results from measurements across four seasons, Atmospheric Chem. Phys., 22, 371–393, https://doi.org/10.5194/acp-22-371-2022,

- 2022.
- Kurtenbach, R., Becker, K. H., Gomes, J. A. G., Kleffmann, J., Lörzer, J. C., Spittler, M., Wiesen, P., Ackermann, R., Geyer, A., and Platt, U.: Investigations of emissions and heterogeneous formation of HONO in a road traffic tunnel, Atmos. Environ., 35, 3385–3394, https://doi.org/10.1016/S1352-2310(01)00138-8, 2001.
- Liu, C., Wang, H., Li, L., Chen, X., Lu, X., and Fan, S.: Impacts of sea-land breeze on the coastal ozone in the Pearl River Delta, China, J. Environ. Sci., https://doi.org/10.1016/j.jes.2025.08.037, 2025.
- Liu, Y., Lu, K., Li, X., Dong, H., Tan, Z., Wang, H., Zou, Q., Wu, Y., Zeng, L., Hu, M., Min, K.-E., Kecorius, S., Wiedensohler, A., and Zhang, Y.: A Comprehensive Model Test of the HONO Sources Constrained to Field Measurements at Rural North China Plain, Environ. Sci. Technol., 53, 3517–3525, https://doi.org/10.1021/acs.est.8b06367, 2019a.
- Liu, Y., Nie, W., Xu, Z., Wang, T., Wang, R., Li, Y., Wang, L., Chi, X., and Ding, A.: Semi-quantitative understanding of source contribution to nitrous acid (HONO) based on 1 year of continuous observation at the SORPES station in eastern China, Atmospheric Chem. Phys., 19, 13289–13308, https://doi.org/10.5194/acp-19-13289-2019, 2019b.
- Ma, X., Tan, Z., Lu, K., Yang, X., Chen, X., Wang, H., Chen, S., Fang, X., Li, S., Li, X., Liu, J., Liu, Y., Lou, S., Qiu, W., Wang, H., Zeng, L., and Zhang, Y.: OH and HO2 radical chemistry at a suburban site during the EXPLORE-YRD campaign in 2018, Atmospheric Chem. Phys., 22, 7005–7028, https://doi.org/10.5194/acp-22-7005-2022, 2022.
- Wang, L., Chai, J., Gaubert, B., and Huang, Y.: A review of measurements and model simulations of atmospheric nitrous acid, Atmos. Environ., 347, 121094, https://doi.org/10.1016/j.atmosenv.2025.121094, 2025.
- Wang, X., Yi, W., Lv, Z., Deng, F., Zheng, S., Xu, H., Zhao, J., Liu, H., and He, K.: Ship emissions around China under gradually promoted control policies from 2016 to 2019, Atmospheric Chem. Phys., 21, 13835–13853, https://doi.org/10.5194/acp-21-13835-2021, 2021.
- Ye, C., Zhou, X., Pu, D., Stutz, J., Festa, J., Spolaor, M., Tsai, C., Cantrell, C., Mauldin, R. L., Campos, T., Weinheimer, A., Hornbrook, R. S., Apel, E. C., Guenther, A., Kaser, L., Yuan, B., Karl, T., Haggerty, J., Hall, S., Ullmann, K., Smith, J. N., Ortega, J., and Knote, C.: Rapid cycling of reactive nitrogen in the marine boundary layer, Nature, 532, 489–491, https://doi.org/10.1038/nature17195, 2016.
- Zaveri, R. A., Easter, R. C., Fast, J. D., and Peters, L. K.: Model for Simulating Aerosol Interactions and Chemistry (MOSAIC), J. Geophys. Res., 113, D13204, https://doi.org/10.1029/2007JD008782, 2008.
- Zhang, H., Ren, C., Zhou, X., Tang, K., Liu, Y., Liu, T., Wang, J., Chi, X., Li, M., Li, N., Huang, X., and Ding, A.: Improving HONO Simulations and Evaluating Its Impacts on Secondary Pollution in the Yangtze River Delta Region, China, J. Geophys. Res. Atmospheres, 129, e2024JD041052, https://doi.org/10.1029/2024JD041052, 2024.
- Zhang, J., Lian, C., Wang, W., Ge, M., Guo, Y., Ran, H., Zhang, Y., Zheng, F., Fan, X.,

- Yan, C., Daellenbach, K. R., Liu, Y., Kulmala, M., and An, J.: Amplified role of potential HONO sources in O3 formation in North China Plain during autumn haze aggravating processes, Atmospheric Chem. Phys., 22, 3275–3302, https://doi.org/10.5194/acp-22-3275-2022, 2022.
- Zhang, L., Wang, T., Zhang, Q., Zheng, J., Xu, Z., and Lv, M.: Potential sources of nitrous acid (HONO) and their impacts on ozone: A WRF-Chem study in a polluted subtropical region, J. Geophys. Res. Atmospheres, 121, 3645–3662, https://doi.org/10.1002/2015JD024468, 2016.
- Zhang, S., Sarwar, G., Xing, J., Chu, B., Xue, C., Sarav, A., Ding, D., Zheng, H., Mu, Y., Duan, F., Ma, T., and He, H.: Improving the representation of HONO chemistry in CMAQ and examining its impact on haze over China, Atmospheric Chem. Phys., 21, 15809–15826, https://doi.org/10.5194/acp-21-15809-2021, 2021.

---

## Author Comment (AC3)

**Response to Reviewer #3**

Dear Editor and Reviewer:

We greatly appreciate your consideration and the reviewer's insightful and constructive comments on the manuscript "HONO Formation Mechanisms and Impacts on Ambient Oxidants in Coastal Regions of Fujian, China" (egusphere-2025-2630). We have carefully revised the manuscript to address all the comments described below. Reviewer comments are shown in black. Our responses are shown in blue. The revised texts are shown in red.

Zhang et al. present a comprehensive analysis of the production and loss mechanisms of HONO, incorporating the updated mechanism into the WRF-Chem model over the coastal region of Fujian, southeastern China. Their measurements, supported by modeling, quantify unusually elevated daytime HONO levels. Through a series of sensitivity simulations, the study further examines the impact of shipping emissions on HONO, as well as the contributions of HONO to OH radical production and O3 formation. Overall, the manuscript is well written and recommended for publication in ACP. Some specific comments are:

**Response:** We thank you for the comments. Based on your helpful and insightful comments, we have revised our manuscript, and the point-by-point responses to the specific comments were given subsequently. We sincerely hope these revisions could address your concerns.

1. Lines 16-18: Briefly mention why HONO concentration is lowest around noon.

**Response:** Thank you. We have rephased this sentence as you suggested.

Revisions in Abstract:

Previous studies have mainly focused on investigating the chemical fate of HONO in polluted urban areas of China and found a general diurnal variation featuring the minimum concentration around noon due to the fast self-photodissociation.

2. In Introduction discuss HONO patterns outside China and cite studies reporting any unusual trends elsewhere as well.

**Response:** Thank you for this insightful suggestion. We acknowledge that the current introduction mainly focuses on studies within China. To provide a broader international context, we have revised the introduction to include a discussion on diurnal pattern of HONO concentrations from other regions worldwide.

**Revisions in Section 1:**

Observational studies conducted in polluted urban areas of China indicated that lower concentrations of HONO typically occurred around noon (Fu et al., 2019; Song et al., 2023; Wang et al., 2025; Zhang et al., 2021). The similar diurnal pattern of HONO concentrations was also reported by measurements in urban areas of Italy, South Korea, Japan, and the United States (Acker et al., 2006; Kim et al., 2024; Nakashima et al., 2017; Stutz et al., 2010).

However, several in-situ observations in typical coastal regions such as Cyprus and Cape Verde revealed an inverse diurnal variation of HONO with higher concentrations occurring at noon (Crilley et al., 2021; Jiang et al., 2023; Meusel et al., 2016). In China, Zhong et al. (2023) reported that their measurement campaign in Qingdao, a coastal city adjacent to the Yellow Sea, identified an unexpected diurnal peak in HONO concentrations at 12:00 local time (UTC+8).

3. Lines 37-39: It will be informative to include reaction rates/photolysis coefficients with references for all reactions discussed in the manuscript.

**Response:** Thanks for your constructive comment. The gaseous and photolysis reactions listed in the manuscript are fundamental atmospheric reactions that are part of the standard SAPRC99 mechanism used in our simulations. To maintain the draft's focus and brevity, we did not list these standard rate constants individually. The rate constants of HONO from muti-phase reactions are shown in Table 2. In the revised manuscript, we have explicitly stated that these rates are standard within the SAPRC99 mechanism and provide a clear reference to the original work for readers seeking detailed information.

**Revisions in Section 1:**

Concurrently, RO2 can react with NO to produce NO2 (R11), a reaction that reduces ozone (O3) titration (R12) while providing NO2 for the subsequent formation of O3 through reactions R13 and R14. The associated reactive rate constants of these listed gaseous and photolysis reactions are well documented in Carter et al. (2000).

4. Line 39: In reaction R3, replace v by Greek letter /nu, if possible.

**Response:** Thanks for your careful reminder. We have corrected all photolysis symbols "hv" to "hv" throughout the revised manuscript.

5. Line 44: Probably authors meant Additionally and not Adaptationally?

**Response:** Thanks for your careful reminder. Yes, it is a typo and has been corrected as follows.

**Revisions in Section 1:**

Additionally, OH radicals can degrade volatile organic compounds (VOCs), ...

6. Lines 59-60: Mention the major reactions contributing to lower HONO concentrations in the noon time in polluted urban environments, as discussed in the previous studies.

**Response:** Thank you for this comment. We have rephased the related texts as you suggested in the revised manuscript.

**Revisions in Section 1:**

Observational studies conducted in polluted urban areas of China indicated that lower concentrations of HONO typically occurred around noon (Fu et al., 2019; Song et al., 2023; Wang et al., 2025; Zhang et al., 2021). The similar diurnal pattern of HONO concentrations was also reported by measurements in urban areas of Italy, South Korea, Japan, and the United States (Acker et al., 2006; Kim et al., 2024; Nakashima et al., 2017; Stutz et al., 2010). These lower HONO concentrations could be attributed to the fast photo-dissociation at noon.

7. Lines 60-61: Please add the reference for the study reporting inverse HONO diurnal variation; if it is Zhong et al. (2023), cite it here as well.

**Response:** Thank you for your careful reminder. We have added the necessary citations regarding the inverse diurnal pattern of coastal HONO concentrations as you suggested.

**Revisions in Section 1:**

However, several in-situ observations in typical coastal regions such as Cyprus and Cape Verde revealed an inverse diurnal variation of HONO with higher concentrations occurring at noon (Crilley et al., 2021; Jiang et al., 2023; Meusel et al., 2016). In China, Zhong et al. (2023) reported that their measurement campaign in Qingdao, a coastal city adjacent to the Yellow Sea, identified an unexpected diurnal peak in HONO concentrations at 12:00 local time (UTC+8).

8. Lines 61-65: Discuss Zhong et al.'s suggested reasons for the unexpected HONO diurnal peak.

**Response:** Thank you for this comment. We have rephrased the relevant texts in the revised manuscript as you suggested.

**Revisions in Section 1:**

In China, Zhong et al. (2023) reported that their measurement campaign in Qingdao, a coastal city adjacent to the Yellow Sea, identified an unexpected diurnal peak in HONO concentrations at 12:00 local time (UTC+8). Using the observation-based model (OBM), their study suggested that the higher HONO concentrations at noon was likely attributed to an unidentified marine source.

9. Lines 64-65: It would be worth discussing in conclusions - the key differences in HONO formation and loss mechanisms between coastal and urban polluted regions. **Response:** Thank you for this constructive suggestion. In Section 3.3.1, we have made some comparisons in chemical fate of HONO between coastal areas and inland regions. We agree with you that systematically summarize and contrast the key HONO formation and loss mechanisms between our coastal study and those typically found in polluted inland urban regions in the conclusion section will strengthen the paper. We have revised the related texts as follows.

**Revisions in Section 4:**

Our study highlights that while the primary daytime loss mechanism for HONO is self-photodissociation in both coastal and highly polluted inland urban areas, their formation mechanisms exhibit significant discrepancies. Previous studies in inland areas often identify the heterogeneous uptake of NO2 on ground surfaces as the dominant source with the maximum contribution up to 86% (Zhang et al., 2024). Our findings in coastal Fujian reveal that the heterogeneous uptake on the ground surface and photo-oxidation of NOx were found to be equally crucial contributors. Furthermore, the contribution from heterogeneous uptake of NO2 on aerosol surfaces, which can be significant in a typical urban environment, was found to be negligible (1-2%) in coastal areas due to much lower aerosol concentrations.

10. Lines 68-69: Please mention the typical number / quantitative of HONO emissions from shipping activities.

**Response:** Thanks for your careful reminder. We have added the reported ratio of HONO to NOx emissions measured by Sun et al. (2020) in the revised manuscript as follows.

**Revisions in Section 1:**

Furthermore, HONO emissions have been acknowledged as a result of shipping activities (Ke et al., 2025; Sun et al., 2020). Sun et al. (2020) estimated that the ratio of HONO to NOx emissions was 0.51% based on specific measurements for ship plumes.

11. Lines 85-86: Do the authors consider this unusual daytime high HONO pattern in coastal regions to be seasonal, or the results valid across all seasons? Additionally, please discuss the applicability of these findings to other coastal regions worldwide in the conclusions.

**Response:** Thank you for this comment. This is a very valuable point. Our study was conducted in May 2024, a period with strong solar radiation in the late spring. We speculate that this photochemically-driven daytime peak pattern would be similarly prominent in summer but may be less pronounced in winter due to weaker solar radiation. More observational evidence should be collected for further investigation in the future. On the other hand, we agree that our findings could be applicable to other mid-latitude coastal regions with similar NOx sources and meteorological conditions. We have added a discussion to the conclusion section on the potential seasonality of our findings and their broader applicability to other coastal regions worldwide.

**Revisions in Section 4:**

This study identifies photochemical-driven processes as the dominant driver of the midday HONO peak in May, a late spring period characterized by abundant solar radiation. Given a more intense solar radiation in summer, it is possible that the daytime HONO formation would be even more pronounced. Whereas in winter, the contribution from these photochemical pathways would likely diminish, potentially leading to a less distinct diurnal pattern where other sources, such as direct emissions, could become relatively more important. Furthermore, the mechanisms discussed in this study might be applicable to other mid-latitude coastal regions worldwide that have similar conditions including NOx precursor sources from shipping emissions and abundant solar radiation. Coastal areas such as the Mediterranean region, the coast of California, and monsoon zones in East Asia may therefore experience similar HONO diurnal variations. To shed more light on coastal HONO chemistry, future long-term, multi-seasonal measurement campaigns in a wider variety of coastal environments are still in need.

12. Lines 102-103: It would be helpful if the authors could further elaborate the statement that the simulation was conducted in a seven-day loop to avoid systemic biases.

**Response:** Thank you for pointing this out. The seven-day loop cycle we used is refer to manually restarting the WRF-Chem model every seven days. The objective this process is to reduce the long time accumulation of biases of numerical computation. To clarify this point, we have revised the associated texts as follows.

**Revisions in Section 2.2:**

The entire WRF-Chem simulation was manually restarted every seven days to reduce the influence of the accumulation of biases from the numerical computation.

13. Lines 135-137: Are the daytime gamma values assumed or based on reported studies? Please provide relevant citations.

**Response:** Thank you for your careful reminder. The maximum uptake coefficient  $\gamma$  was chosen based on previous modeling studies. We have revised the related texts to attach the source reference.

**Revisions in Section 2.3:**

The base nighttime uptake coefficients of  $NO_2$  were set to  $8 \times 10^{-6}$  for the ground surface and  $4 \times 10^{-6}$  for aerosol surfaces. During the daytime, these values were dynamically increased with solar radiation using a linear equation, reaching their maximums of  $6 \times 10^{-5}$  and  $1 \times 10^{-3}$ , respectively (Liu et al., 2019).

14. Lines 174-176: It would be nice to mention the modelled HONO concentrations as well for Beijing to assess how well they match observations.

Response: Thanks for your insightful comment. We would like to clarify this point. The mention of Beijing on lines 174-176 in the original manuscript was intended to provide context by comparing the observed HONO levels at coastal site in Fujian with observed levels from a heavily polluted region in China. However, the modeling domain for this study was focused on southeastern China and did not include Beijing. Therefore, we do not have any simulated HONO concentrations for Beijing from this study to compare with previous observations. We would like to express our gratitude to your constructive suggestion again.

15. Line 185-186: Are the definitions of non-rainy and rainy days based on a specific metric?

**Response:** Thank you for pointing out the need for a precise definition. In our analysis, rainy periods (shaded areas in Fig. 2) were defined as any hour during which the regional meteorological station in Putian recorded non-zero precipitation. We have added this specific metric to the manuscript for clarity as you suggested.

**Revisions in Section 3.1:**

Figure S2 exhibits the diurnal variations of gaseous air pollutants and meteorological parameters. Based on observations of the regional meteorological station in Putian, we categorized any hour with non-zero recorded precipitation into rainy days.

16. Lines 190-191: It would be informative to include standard deviations in Fig. S2, particularly for the HONO, NO2, and O3 plots.

**Response:** Thanks for this valuable comment. We have adjusted Figure S2 as you suggested to include plots of standard deviations for meteorological parameters and air pollutants.

**Revisions in Figure S2:**

17. Lines 193-194: Discuss the possible reasons for high O3 at 4 pm despite low meteorological parameters?

Response: Thank you for pointing this out. The lag between the peak in solar radiation around 14:00 and the peak in O3 concentration at 16:00 could be attributed to the favorable atmospheric oxidizing capacity resulting from the

simultaneous high HONO levels. O3 is a secondary pollutant whose formation from the photochemical oxidation of VOCs and NOx is a cumulative process that takes several hours. The chemical production rate of O3 remains high even as solar radiation begins to decline, leading to the O3 peak occurring later in the afternoon. We has added this discussion to the revised manuscript.

**Revisions in Section 3.1:**

Despite the decrease in air temperature and radiation after 14:00, the O3 concentration increased until it reached a daily maximum of 64.8 ppbv at 16:00. The lag of O3 peak concentration could be attributed to the strengthened AOC conditions resulting from the simultaneous high HONO levels. The chemical production of O3 is a cumulative process that takes several hours so that O3 levels could remain high even as solar radiation begins to decline, leading to the O3 peak occurring later in the afternoon.

- Lines 215-216: Please mention correct figure number.
  Response: Thanks for your kind reminder. The typo has been corrected.
- 19. Figure 3: To avoid confusion, it would be better to label the legends as REV and BASE, respectively, instead of 'Mean model' for both.

**Response:** Thanks for your careful comment. We have adjusted the legends of Figure 3a as you suggested for clarity as follows.

**Revisions in Figure 3a:**

20. Line 224: In Figure 4, NO2 values for BASE and REV appear very similar and closer to each other than to the observations. Does this mean the updates have little

effect on NO2 concentrations? Notably, the 24-hour mean NO2 for BASE seems slightly closer to observations than REV. Please clarify.

**Response:** Thanks for your conducive comments and rigorous attitude to scientific research. You are correct that, as shown in Figure 4, the updates to the HONO chemistry in the REV simulation resulted in little net change to the simulated NO2 concentrations compared to the BASE case. The 24-hour mean NO2 in the REV case is even slightly further from the observations. This phenomenon occurs because the updated HONO chemistry can enhance the atmospheric oxidizing capacity, which has a more direct and significant impact on the formation of secondary pollutants like O3. In contrast, the net effect on NO2 is more complicated and multifaceted, resulting from several competing chemical pathways: while the new HONO sources act as a sink for NO2, the concurrently enhanced OH radical concentrations can also promote the conversion of NO to NO2, and the changes in O3 concentration also influence NO2 levels via the titration effect. These competing processes result in only a small net change in the overall NO2 concentration. Thus, the slight increase in the model bias for NO2 in REV is a reasonable trade-off for the substantial improvements achieved in the simulations of HONO and O3, which were the major objectives of our model updates. To make this point clearer, we have incorporated an additional in the revised manuscript as follows.

**Revisions in Section 3.2:**

The updated HONO chemistry had a limited net impact on NO2 concentrations due to several competing chemical pathways. Specifically, the consumption of NO2 by the newly added HONO formation reactions was largely offset by enhanced NO-to-NO2 conversion from increased OH radicals and altered O3 titration, resulting in only a minor change in the overall NO2 budget.

21. Lines 240-242: It would be helpful to include a supplementary plot of the net production rate (Production – Loss) to show when production dominates.

**Response:** Thanks for this insightful comment. We have added the curve of net HONO production rate (using production rate minus loss rate) in Figure S4 to clearly show the temporal variation of net HONO chemical production as you suggested.

Revisions in Figure S4:

22. Line 347: Along with OH production rates, it would be informative to show OH loss rates too, since NOy, HOy, CO, CH4, and VOCs significantly consume OH.

**Response:** Thank you for this constructive suggestion. We agree that incorporating the various chemical sink pathways for OH radicals would make our analysis more informative. The source-oriented method (SOM) employed in this study is designed for source apportionment by diagnosing the rate changes from different reactive pathways. While the WRF-Chem model already uses a lumped approach to reduce the number of VOC variables, applying the SOM to also diagnose the vast number of OH sink reactions, particularly with the numerous VOC species, would introduce a substantial additional computational and storage burden. Considering that the primary objective of Section 3.4.1 was to quantify the contribution of HONO to primary sources of OH radicals, the utility of rerunning our computationally intensive simulations to calculate OH sinks may be limited in the context of this specific goal. Nevertheless, we believe your suggestion is highly valuable as it motivates us to further investigate the broader importance of the enhanced atmospheric oxidative capacity driven by HONO. For future studies, we will develop targeted modifications to our SOM approach to improve the efficiency of calculating OH sinks, and we intend to incorporate this investigation into our future work. Thank you again for this valuable suggestion.

**References**

Acker, K., Febo, A., Trick, S., Perrino, C., Bruno, P., Wiesen, P., Möller, D., Wieprecht, W., Auel, R., Giusto, M., Geyer, A., Platt, U., and Allegrini, I.: Nitrous acid in the urban area of Rome, Atmos. Environ., 40, 3123–3133, https://doi.org/10.1016/j.atmosenv.2006.01.028, 2006.

Carter, W. P. L.: "Documentation of the SAPRC-99 Chemical Mechanism for VOC Reactivity Assessment," Report to the California Air resources Board, Contracts

- 92-329 and 95-308, 2000.
- Crilley, L. R., Kramer, L. J., Pope, F. D., Reed, C., Lee, J. D., Carpenter, L. J., Hollis, L. D. J., Ball, S. M., and Bloss, W. J.: Is the ocean surface a source of nitrous acid (HONO) in the marine boundary layer?, Atmospheric Chem. Phys., 21, 18213–18225, https://doi.org/10.5194/acp-21-18213-2021, 2021.
- Fu, X., Wang, T., Zhang, L., Li, Q., Wang, Z., Xia, M., Yun, H., Wang, W., Yu, C., Yue, D., Zhou, Y., Zheng, J., and Han, R.: The significant contribution of HONO to secondary pollutants during a severe winter pollution event in southern China, Atmospheric Chem. Phys., 19, 1–14, https://doi.org/10.5194/acp-19-1-2019, 2019.
- Jiang, Y., Hoffmann, E. H., Tilgner, A., Aiyuk, M. B. E., Andersen, S. T., Wen, L., van Pinxteren, M., Shen, H., Xue, L., Wang, W., and Herrmann, H.: Insights Into NOx and HONO Chemistry in the Tropical Marine Boundary Layer at Cape Verde During the MarParCloud Campaign, J. Geophys. Res. Atmospheres, 128, e2023JD038865, https://doi.org/10.1029/2023JD038865, 2023.
- Ke, J., Yang, X., Lu, K., Fu, M., Wang, Y., Yin, H., and Ding, Y.: Overlooked Underestimation of Mobile Sources Posing a Pronounced Imbalance in the HONO Budget, Environ. Sci. Technol., 59, 5875–5877, https://doi.org/10.1021/acs.est.5c02684, 2025.
- Kim, K., Han, K. M., Song, C. H., Lee, H., Beardsley, R., Yu, J., Yarwood, G., Koo, B., Madalipay, J., Woo, J.-H., and Cho, S.: An investigation into atmospheric nitrous acid (HONO) processes in South Korea, Atmospheric Chem. Phys., 24, 12575–12593, https://doi.org/10.5194/acp-24-12575-2024, 2024.
- Liu, Y., Lu, K., Li, X., Dong, H., Tan, Z., Wang, H., Zou, Q., Wu, Y., Zeng, L., Hu, M., Min, K.-E., Kecorius, S., Wiedensohler, A., and Zhang, Y.: A Comprehensive Model Test of the HONO Sources Constrained to Field Measurements at Rural North China Plain, Environ. Sci. Technol., 53, 3517–3525, https://doi.org/10.1021/acs.est.8b06367, 2019.
- Meusel, H., Kuhn, U., Reiffs, A., Mallik, C., Harder, H., Martinez, M., Schuladen, J., Bohn, B., Parchatka, U., Crowley, J. N., Fischer, H., Tomsche, L., Novelli, A., Hoffmann, T., Janssen, R. H. H., Hartogensis, O., Pikridas, M., Vrekoussis, M., Bourtsoukidis, E., Weber, B., Lelieveld, J., Williams, J., Pöschl, U., Cheng, Y., and Su, H.: Daytime formation of nitrous acid at a coastal remote site in Cyprus indicating a common ground source of atmospheric HONO and NO, Atmospheric Chem. Phys., 16, 14475–14493, https://doi.org/10.5194/acp-16-14475-2016, 2016.
- Nakashima, Y., Sadanaga, Y., Saito, S., Hoshi, J., and Ueno, H.: Contributions of vehicular emissions and secondary formation to nitrous acid concentrations in ambient urban air in Tokyo in the winter, Sci. Total Environ., 592, 178–186, https://doi.org/10.1016/j.scitotenv.2017.03.122, 2017.
- Song, M., Zhao, X., Liu, P., Mu, J., He, G., Zhang, C., Tong, S., Xue, C., Zhao, X., Ge, M., and Mu, Y.: Atmospheric NOx oxidation as major sources for nitrous acid (HONO), Npj Clim. Atmospheric Sci., 6, 30, https://doi.org/10.1038/s41612-023-00357-8, 2023.
- Stutz, J., Oh, H.-J., Whitlow, S. I., Anderson, C., Dibb, J. E., Flynn, J. H., Rappenglück, B., and Lefer, B.: Simultaneous DOAS and mist-chamber IC measurements of

- HONO in Houston, TX, Atmos. Environ., 44, 4090–4098, https://doi.org/10.1016/j.atmosenv.2009.02.003, 2010.
- Sun, L., Chen, T., Jiang, Y., Zhou, Y., Sheng, L., Lin, J., Li, J., Dong, C., Wang, C., Wang, X., Zhang, Q., Wang, W., and Xue, L.: Ship emission of nitrous acid (HONO) and its impacts on the marine atmospheric oxidation chemistry, Sci. Total Environ., 735, 139355, https://doi.org/10.1016/j.scitotenv.2020.139355, 2020.
- Wang, L., Chai, J., Gaubert, B., and Huang, Y.: A review of measurements and model simulations of atmospheric nitrous acid, Atmos. Environ., 347, 121094, https://doi.org/10.1016/j.atmosenv.2025.121094, 2025.
- Zhang, H., Ren, C., Zhou, X., Tang, K., Liu, Y., Liu, T., Wang, J., Chi, X., Li, M., Li, N., Huang, X., and Ding, A.: Improving HONO Simulations and Evaluating Its Impacts on Secondary Pollution in the Yangtze River Delta Region, China, J. Geophys. Res. Atmospheres, 129, e2024JD041052, https://doi.org/10.1029/2024JD041052, 2024.
- Zhang, S., Sarwar, G., Xing, J., Chu, B., Xue, C., Sarav, A., Ding, D., Zheng, H., Mu, Y., Duan, F., Ma, T., and He, H.: Improving the representation of HONO chemistry in CMAQ and examining its impact on haze over China, Atmospheric Chem. Phys., 21, 15809–15826, https://doi.org/10.5194/acp-21-15809-2021, 2021.
- Zhong, X., Shen, H., Zhao, M., Zhang, J., Sun, Y., Liu, Y., Zhang, Y., Shan, Y., Li, H., Mu, J., Yang, Y., Nie, Y., Tang, J., Dong, C., Wang, X., Zhu, Y., Guo, M., Wang, W., and Xue, L.: Nitrous acid budgets in the coastal atmosphere: potential daytime marine sources, Atmospheric Chem. Phys., 23, 14761–14778, https://doi.org/10.5194/acp-23-14761-2023, 2023.